# Inclusive fitness forces of selection in an age-structured population

Mark Roper [1,2✉], Jonathan P. Green[1], Roberto Salguero-Gómez[1,3] & Michael B. Bonsall [1]

Hamilton's force of selection acting against age-specific mortality is constant and maximal prior to the age of first reproduction, before declining to zero at the age of last reproduction. The force of selection acting on age-specific reproduction declines monotonically from birth in a growing or stationary population. Central to these results is the assumption that individuals do not interact with one another. This assumption is violated in social organisms, where an individual's survival and/or reproduction may shape the inclusive fitness of other group members. Yet, it remains unclear how the forces of selection might be modified when inclusive fitness, rather than population growth rate, is considered the appropriate metric for fitness. Here, we derive such inclusive fitness forces of selection, and show that selection on age-specific survival is not always constant before maturity, and can remain above zero in post-reproductive age classes. We also show how the force of selection on age-specific reproduction does not always decline monotonically from birth, but instead depends on the balance of costs and benefits of increasing reproduction to both direct and indirect fitness. Our theoretical framework provides an opportunity to expand our understanding of senescence across social species.

---

[1] Department of Biology, University of Oxford, 11a Mansfield Road, Oxford OX1 3SZ, UK. [2] Institute for Biodiversity and Ecosystem Dynamics, University of Amsterdam, Amsterdam, The Netherlands. [3] Max Planck Institute for Demographic Research, Konrad-Zuse-Straße 1, 18057 Rostock, Germany. ✉email: markroper67@gmail.com

At the demographic level, senescence is defined as the decline in organismal fitness with increasing age[1]. Hamilton[2] provided a mathematical explanation for the seemingly counter-intuitive evolution of senescence: the force of natural selection weakens with age, and so detrimental alleles acting late in life can persist despite their negative effects on fitness[3–6]. Two years prior, Hamilton[7,8] also introduced the concept of inclusive fitness, which has had a profound impact on our understanding of the evolution of social life histories[9–11]. Inclusive fitness quantifies (i) an individual's number of offspring in the absence of social effects and (ii) the effects an individual has on the number of offspring produced by other individuals, weighted by relatedness[7,8]. However, despite proposed verbal mechanisms[12,13], and theoretical studies that have considered either specific social interactions[14–16], or nutritional transfers between individuals[17,18], there is no framework that focuses on (computations of) the forces of selection for both age-specific survival and reproduction using inclusive fitness methodology, for any arbitrarily complex range of social interactions. For this reason, we lack a good understanding of how social interactions influence the evolution of senescence across species exhibiting a diverse array of social systems.

In his original work on the evolution of senescence, Hamilton quantified the forces of selection as the effect on fitness of a hypothetical mutant allele compared to a resident wild-type allele[2]. The phenotypic effect of the allele could be, for example, to increase incrementally the mortality risk at age $x$, reducing survival into the next age class. As a thought experiment, consider how the effects on fitness of a mutant allele that alters mortality risk might differ if this allele invaded in a solitary *vs.* a social species. First, consider an individual of a solitary species. When this individual dies, it loses access to any future reproduction it might have achieved. If a mutant allele arises in this population that increases the risk of dying at a certain age, say $x$, then the force of selection that acts against the allele is proportional to the expectation of residual reproduction that the individual may have realised if it survived for longer[2]. Now, imagine instead a social species in which individuals within a group influence one another's survival and reproduction, for example, through the provision of alloparental care or through competition for limiting resources. For an individual, death means the loss of any future reproduction, just as in the solitary case. However, in social species, an individual's death may also alter the survival and reproduction of other individuals[12,13]. For instance, the death of an individual providing alloparental care may lead to a reduction in breeder productivity. Alternatively, where there is competition within groups for resources, the death of an individual may release resources that other group members may use for survival and reproduction[12]. If individuals within a group are related, then these effects will be under kin selection. For example, an increase in mortality late in life can be adaptive if relatives stand to benefit from the death of a focal individual[14,19–24]. On the other hand, mortality may be more strongly selected against if individuals can transfer beneficial resources to others[15,17,25]. When the death and reproduction of a focal individual not only impacts its own fitness, but also the fitness of relatives, the force of selection acting on a mutant allele at age $x$ must also consider these complex social effects. Incorporating these complex social effects into the forces of selection may generate novel predictions for life history evolution in social organisms.

In his seminal work[2], Hamilton noted (pg. 23) that "*the inclusive fitness of an individual is maximized by continually acting in ways that cause increases in its inclusive reproductive value*". Yet, when computing the forces of selection, he considered the standard population growth rate, $r$, as the measure of fitness, and did not consider indirect contributions of individuals

to the fitness of others. Hamilton did, however, suggest a potential mechanism of 'sibling replacement', by which the force of selection acting against age-specific mortality might increase as juveniles age closer to maturity[2]. In earlier work, Medawar[3] and Williams[4] also hinted at the importance of the indirect actions of selection when there is any high degree of social organization[3], and, in the latter case, the specific relevance of parental care in explaining post-reproductive life[4]. Indeed, if offspring survival to reproductive maturity is dependent on parental or grandparental care, then forces of selection on adult age-specific mortality and fertility may be modified[15,16]. Selection for menopause and continued post-reproductive survival may be favoured if individuals can benefit from the survival of juveniles, and may be under stronger selection in conditions of reproductive competition between related females[26–28]. Grandparental care is, in fact, a particular case of alloparental care. Yet, despite evidence that alloparents might reduce senescence rates[29], or increase the lifespans[30], of breeders in cooperative breeding systems, evolutionary models have yet to consider how alloparental care may alter forces of selection.

Many social organisms live in groups with overlapping generations[31]. Overlapping generations may, in fact, be of fundamental importance to the evolution of cooperation[32], and many theoretical studies have examined conditions for the invasion of a cooperative allele under kin selection in age-structured populations[21,32–36]. Such studies have revealed that when considering the evolutionary fate of an allele in age-structured populations, effects on the fitness of individuals, as well as being weighted by relatedness, must also be weighted by reproductive value[33–35]. Social behaviours that result in changes to survival must be weighted by the reproductive value of the age class following that to which an individual belongs (i.e. $x + 1$)[33]. If interactions instead involve effects on reproduction, the behaviour is weighted by the reproductive value of newborns, which is often assumed to be equal to one[37]. Conventional reproductive value, as defined by Fisher[38], does not, however, capture the genetic contributions individual's make through indirect contributions to the fitness of others. Such a quantity, referred to informally by Hamilton[2] as *inclusive reproductive value* (IRV), is yet to be fully defined.

Many studies have focused on how kin selection determines the evolutionary fate of an allele with background age-specific mortality and fertility schedules[21,32–37]. Bourke[12], however, instead considered how such individual mortality and fertility schedules may affect the fitness of others in social organisms, and in turn introduced the potential for effects of kin interactions on forces of selection. In addition to Bourke's hypotheses[12], simulation studies have investigated the relationship between limited dispersal and the evolution of shorter lifespan[22], but often assume some form of senescence that has already evolved in the population. Nutritional transfer models have sought to extend Hamilton's forces of selection for age-specific mortality beyond the individual[17,18], but do not consider the force of selection for reproduction. A spatially explicit model has investigated the interaction between kin competition and forces of selection[14], yet was limited to parent-offspring conflict. Currently, there is no framework that allows for any arbitrarily complex range of social interactions between age classes, from which inclusive fitness forces of selection for both age-specific survival and reproduction can be quantified. The goal of this work is to develop the foundations for such a framework.

Here, we develop a model to quantify inclusive fitness forces of selection on age-specific survival and reproduction in a population where individuals exhibit behaviour(s) that alter another individual's survival and reproduction. We focus here on the effects of cooperative interactions between individuals

and the corresponding forces of selection, but note that our model also has scope to consider other scenarios, such as cases of harm (see 'Discussion'). Our main focus is to develop analytical solutions that describe the general components of age-specific inclusive fitness forces of selection, and then provide specific numerical examples to illustrate the application of the theory. Our framework can, however, consider any arbitrarily complex direction and magnitude of social interactions between age classes. We do not consider the invasion of a cooperative allele, but instead assume that individuals in our hypothetical social species have evolved some set of age-specific social interactions that are stable in the population (see 'Discussion'). Using an infinite island framework to describe such a resident social population[14,21,26,27,32,39–45], we explore the fate of a mutant allele in two scenarios. First, we consider an allele that alters the survival rate from age $x$ to age $x + 1$ and, second, we consider an allele that alters the rate of reproduction at age $x$. We derive age-specific inclusive fitness forces of selection acting on these mutant alleles for each age $x$ from 1 to some maximum age, $\omega$. We show the stark difference in computing forces of selection via Hamilton's methodology[2], not considering transfers, and our methodology, by exploring the applicability of our framework to different social structures: (i) the grandmother hypothesis: post-reproductive individuals aiding juvenile survival and (ii) cooperative breeding: juveniles aiding reproduction by adults. We conclude by discussing the implications, limitations, and possible extensions to our work.

## Results

**Inclusive fitness model**. We consider a population divided into an infinite number of patches, and model the population dynamics of a focal patch. Each patch, which could be conceptualised, for example, as a territory, contains discrete groups of exactly $N$ individuals that are, for simplicity, haploid and asexual. A proportion of offspring of age class 1 disperse to other patches; when this proportion is between 0 and 1, patches comprise both kin and non-kin, leading to variation in pairwise relatedness. Variation in pairwise relatedness is also driven by the number of individuals on the patch, and age-specific rates of reproduction and survival (see 'Methods'). Offspring that establish onto a patch at age 1 can survive until some maximum age, $\omega$, at which point they die. Time proceeds in a series of discrete breeding seasons, during which each of the $N$ individuals on a patch have a probability of surviving to the next breeding season, creating overlapping generations. We assume that the absolute value of individual reproduction is large enough at all ages so that no position on any patch is vacant at the start of each breeding season (i.e. a stationary population). When deriving inclusive fitness forces of selection, we focus on a focal individual aged $x$ and seek to identify the inclusive fitness consequences of an allele that changes the survival probability or the rate of reproduction of the individual. We assume that the mutation is sufficiently rare, so that the other individuals, besides the focal, reproduce and survive at population average levels that are calculated by weighting each age-specific probability of survival or rate of reproduction by the stationary age distribution of the respective age class. We discuss the potential consequences of our main assumptions, specifically (i) a stationary population, (ii) lack of environmental or demographic stochasticity, and (iii) non-evolving social traits in the 'Discussion'.

**Transfers**. Fundamental to this model is the concept of 'transfers'. Elsewhere, transfers have been described as directional contributions of resources from one individual to another, often in the form of nutritional benefit(s)[17,18]. Here, instead, we

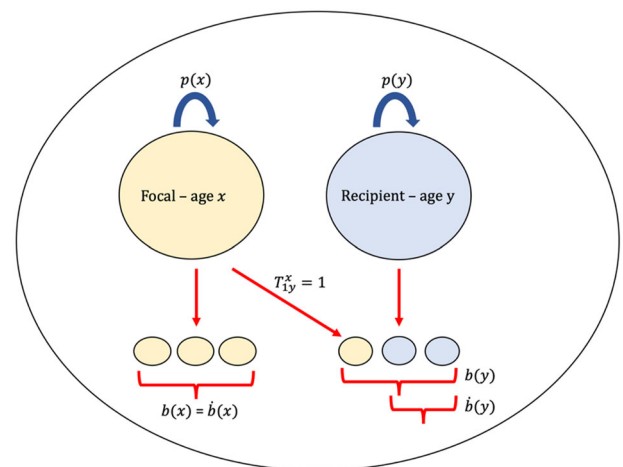

**Fig. 1 An example of a genetic offspring transfer between two individuals using inclusive fitness methodology.** To illustrate transfers, we consider a patch with two individuals, one of age $x$ and the other of age $y$. The individual aged $x$ has $b(x)$ offspring, survives with probability $p(x)$, and receives no social transfers from other individuals in the population when aged $x$. We imagine a social behaviour exists whereby the individual aged $x$ contributes to the reproduction of individuals aged $y$. In this scenario, the individual aged $y$ has $b(y)$ offspring in the current breeding season, but one of these offspring is due to the transfer from the focal individual aged $x$. Following inclusive fitness logic, the offspring produced due to the social behaviour of the individual aged $x$ is stripped from the inclusive fitness of the individual aged $y$, leaving $\dot{b}(y)$ as their inclusive fitness contribution to age class 1. The inclusive fitness contribution of the focal individual aged $x$ to age class 1 is $\dot{b}(x) + T_{1y}^{x}$, where $T_{1y}^{x}$ is weighted by the relatedness of an individual aged $x$ to the offspring it helped to produce ($\hat{r}(x)$). In the figure we assume $\hat{r}(x) = 1$.

define transfers between age classes as the per capita net contribution of all helpful or harmful social behaviours. Transfers are in the currency of genetic offspring equivalents, the same currency as survival and reproduction. Essentially, transfers, like survival and reproduction, represent (fractional) copies of an individual in the next breeding season. Individuals may receive transfers to their survival and reproduction from the other $N-1$ individuals on their patch, and may themselves contribute transfers via the survival and reproduction of the $N-1$ conspecifics on the patch. We denote transfers between individuals according to their age classes as $T_{yz}^{x}$ (Fig. 1; Table 1). The superscript, $x$, denotes the age class of the actor. The subscript then denotes the age classes of the recipient, $z$, and whether the contribution is to the survival or reproduction of the recipient (defined by $y$). If $y = 1$ ($T_{1z}^{x}$), this represents an individual in age class $x$'s indirect reproduction via age class $z$. Alternatively, when $y = z + 1$ ($T_{z+1,z}^{x}$), this represents an individual in age class $x$'s contribution to the survival of age class $z$ individuals. The whole term, $T_{yz}^{x}$, then quantifies the per-capita genetic offspring equivalent contribution of individuals aged $x$ via the survival or reproduction of an individual aged $z$. Note, as $T_{yz}^{x}$ represents genetic offspring equivalents, it is implicitly weighted by relatedness. For numerical solutions, we derive relatedness explicitly (see 'Methods'). Importantly, as mentioned earlier, we assume the population has evolved some stable strategy set of transfers between age classes of individuals. Our infinite island modelling approach, often used in models that consider the invasion of a cooperative allele, allows for future work to investigate the co-evolution of specific helping behaviours and 'senescent' alleles.

**Table 1 List of parameters and definitions used in (i) defining the resident population and (ii) quantifying inclusive fitness forces of selection in the model.**

| Parameter | Definition |
|---|---|
| *Defining the resident population* | |
| $N$ | The number of individuals on a patch, fixed. |
| $T_{yz}^x$ | The number of class $y$ offspring contributed per capita through social transfers by age class $x$ to age class $z$, through the survival ($y = z + 1$) or reproduction ($y = 1$) of age class $z$ individuals. The term is weighted by relatedness of age class $x$ individuals to age class $z$ individuals. |
| $p(x)$ | The survival probability of age class $x$ to age $x + 1$. Can be decomposed into the survival due to the genotype of an individual aged $x$, and the help it receives towards survival from other individuals on the patch ($p(x) = \dot{p}(x) + \sum_y T_{x+1,x}^y$). |
| $\dot{p}(x)$ | The number of class-$x + 1$ offspring of an age class $x$ through survival, stripped of all social effects. |
| $\bar{p}$ | Average survival rate of individuals, determined by weighting age-specific survival $p(x)$ by the stationary age distribution ($s(x)$). |
| $s(x)$ | The stationary age distribution of age class $x$. |
| $b(x)$ | The reproductive rate of age class $x$. Can be decomposed into the reproduction due to the genotype of an individual aged $x$, and the help it receives towards reproduction from other individuals on the patch ($b(x) = \dot{b}(x) + \sum_y T_{1,x}^y$). |
| $\dot{b}(x)$ | The reproductive rate of age class $x$, stripped of all social effects. |
| $\bar{b}$ | Average reproductive rate of individuals, determined by weighting age-specific reproduction ($b(x)$) by the stationary age distribution ($s(x)$). |
| $d$ | The dispersal rate of age class 1 offspring, fixed. |
| $c$ | The cost of dispersal of age class 1 offspring, fixed. |
| $g(x)$ | The probability of establishment of a class 1 offspring on the patch with an age $x$ individual. |
| $\bar{g}$ | Average probability of establishment of a class 1 offspring on a random patch in the population. |
| $\dot{F}(x)$ | The number of class-1 offspring of a class $x$ individual, produced directly ($\dot{b}(x)$) and through the reproduction of others ($\sum_z T_{1,z}^x$). |
| *Quantifying forces of selection* | |
| $\hat{r}(x)$ | The average relatedness of a breeding individual aged $x$ to another random breeder on the same patch. |
| $h(x)/\dot{h}(x)$ | The proportion of offspring after dispersal at the local patch that are the offspring (not partitioned into inclusive fitness contributions) of a focal individual aged $x$/the proportion of offspring after dispersal at the local patch that are produced due to the genotype (i.e. $\dot{b}(x) + \sum_y T_{1,x}^y$) of a focal individual aged $x$. |
| $k(x)/\dot{k}(x)$ | The proportion of offspring after dispersal at the local patch that are the offspring (not partitioned into inclusive fitness contributions) of other individuals on the patch/the proportion of offspring that are born due to the genotypes of other individuals on the patch. |

**The resident population.** The life cycle of a species can often be described in matrix format[46]. For example, consider a standard age-classified population projection matrix[47] ($A$), where the element $a_{yx}$ describes the contribution of age class $x$ (column) to the production of age class $y$ (row) individuals over a particular time period (e.g. one year). The elements along the sub-diagonal of the matrix represent age-specific survival probabilities, whilst the elements along the top row represent age-specific reproduction. In a monomorphic population, the dominant eigenvalue associated with the matrix $A$ is the population growth rate[35]. The approach we develop here is to assume that fractions of the contributions between age classes described in $A$ are due to social interactions (transfers). These transfers are redistributed among age classes, translating the form of $A$ so that it describes inclusive fitness contributions. We call this translated matrix $W$, and its dominant eigenvalue then represents an inclusive fitness modified version of the population growth rate.

To translate the elements in $A$ into inclusive fitness contributions, a series of key considerations must be made. Specifically, following Hamilton's definition of inclusive fitness[7,8], we must (i) exclude the fraction of the class-$y$ offspring of a focal class-$x$ individual that are born or survive as a consequence of the social environment (the help or harm of other individuals), and (ii) augment the total production of class-$y$ offspring from all other age classes, including other individuals in age class $x$, that are born or survive due to the social contributions of a focal class-$x$ individual. These latter offspring contributions are weighted by the coefficient of relatedness between an individual of age class $x$ and the class-$y$ offspring of the recipient class[7,8]. We use the demography detailed in the infinite island structure of our model to perform the inclusive fitness methodology expressed in (i) and (ii). For example, an individual aged $x$ survives with probability $p(x)$ and has a rate of reproduction $b(x)$. As stated in (i) we must exclude the fraction of these rates of survival and reproduction that are due to social interactions. This leaves $\dot{p}(x)$ and $\dot{b}(x)$, with

dot notation representing the effect of a focal individual's own genotype on its own survival or rate of reproduction, i.e. direct fitness, and ensuring no offspring are double counted[48,49]. For (ii), we then augment the focal individual's fitness with the genetic representation it gains through transfers to other age classes (all $T_{yz}^x$). In doing so, we can the translate the elements of $A$ into the inclusive fitness matrix ($W$) of the resident population as:

$$w_{yx} = \begin{cases} \dot{p}(x) + T_{x+1,x}^x & \text{if } y = x + 1 \\ \dot{F}(x)D & \text{if } y = 1 \\ 0 \text{ OR } T_{yz}^x & \text{if } y = z + 1 \end{cases} \quad (1)$$

where

$$\dot{p}(x) = p(x) - \sum_z T_{x+1,x}^z \quad (2)$$

$$\dot{b}(x) = b(x) - \sum_z T_{1,x}^z \quad (3)$$

$$\dot{F}(x) = \dot{b}(x) + \sum_z T_{1,z}^x \quad (4)$$

and

$$D = \left[ (1 - d)g(x) + (1 - c)d\bar{g} \right] \quad (5)$$

Each row of (1) represents the inclusive fitness contributions from an individual aged $x$ in a given breeding season. The top row of (1) represents the sum of individual survival (the focal individual produces $\dot{p}(x)$ copies of themselves next year stripped of social effects (2)), and the contributions to the survival of same aged individuals ($T_{x+1,x}^x$). The second row of (1) represents contributions to the offspring age class. The total number of genetic offspring equivalents for a focal individual aged $x$ is the sum of their offspring, stripped of social effects (3), and their contributions to the reproduction of others, summed across age classes to equal $\sum_z T_{1,z}^x$. As the newborn offspring class can

disperse, we need to consider the fate of both dispersing and non-dispersing offspring to accurately quantify $w_{1x}$ (5). A proportion $d$ disperse, and a proportion $1 - d$ remain at their natal patch. A fraction, $c$, of the dispersing offspring die representing a cost of dispersal. Surviving, dispersed offspring are evenly distributed among all sites and compete (fair lottery) for sites freed by adults that die in the current breeding season. Asymmetric competition is assumed such that juveniles do not displace resident adults. Juveniles that do not gain a breeding position on a patch die. Offspring of an individual aged $x$ establish on their natal patch with probability $g(x)$, and with probability $\bar{g}$ on a different, random, patch in the population. The term $\bar{g}$ represents the average probability of establishment for an offspring on another patch, which considers the expected level of the survival of patch members according to the expected age distribution of patch members. The final row of (1) considers the social effects of a focal individual aged $x$ on the survival of individuals in other age classes.

In summary, our approach assumes that there are fractions of age-specific survival and reproduction that are due to the social environment (which could equal zero), and that these fractions are distributed to other individuals across age classes. This ensures that the elements in $\mathbf{W}$ are inclusive fitness contributions between age classes. If there are no transfers between the individuals on a patch, the elements of $\mathbf{W}$ described in (1) simplify to an $N>1$ version of Ronce and Promislow's[14] kin competition model with limited dispersal. The forces of selection they derive are not influenced by of multiple individuals on a patch occupying the same patch, if there are no transfers. An additional simplification of full dispersal (no offspring stay at their natal patch), renders the elements of $\mathbf{W}$ described in (1) equivalent to a panmictic population with vital rates $p(x)$ and $b(x)$. In this case, Hamilton's forces of selection can be computed[2].

**A mutant allele**. Now we have defined the resident population, our analysis considers the effect on inclusive fitness of a hypothetical mutation that alters the survival rate or rate of reproduction at age $x$ in the resident population. The derivative of the growth rate of the mutant population, $\lambda$, with respect to the phenotypic effect of the mutation, $\delta$, generates the force of selection acting on the mutant allele[2,14,50,51]. We consider mutations of weak effects (small $\delta$) and first-order (i.e. linear) effects of selection[52]. Using this 'sensitivity' approach for an age-structured population[14,35,50–53], the force of selection acting on a mutant allele can be written as:

$$S = \frac{d\lambda}{d\delta}\Big|_{\delta=0} = \sum_x \sum_y \frac{f_x v_y}{\mathbf{f} \cdot \mathbf{v}} \frac{dw_{yx}}{d\delta}\Big|_{\delta=0} \quad (6)$$

Here, $\lambda$, represents an inclusive fitness growth rate of the allele taken as the dominant eigenvalue associated with $\mathbf{W}$. The vectors $\mathbf{f}$ and $\mathbf{v}$ are, respectively, the right and left eigenvectors associated with $\lambda$ (the dominant eigenvalue of $\mathbf{W}$). The term $v_x$ represents the inclusive reproductive value of age class $x$, a modified version of conventional reproductive value[38], which measures the contribution of age class $x$ to the ancestry of future generations through direct and indirect contributions. The term $f_x$ has a slightly less intuitive interpretation. It represents a modified version of the asymptotic frequency of age class $x$, i.e. the stationary age distribution of age class $x$, that accounts for transfers between age classes. Finally, given $w_{yx}$, which represents the class $y$ offspring of a class $x$ individual (genetic offspring equivalents), $dw_{yx}$ then represents the difference in the contribution of a mutant individual aged $x$ to individuals aged $y$ relative to the resident population. Overall, the sign of $S$ predicts the direction

of selection on the mutant allele with respect to the resident population wild-type allele, whilst a larger absolute magnitude of $S$ signals a stronger force of selection[2,14,50].

**The inclusive fitness force of selection on survival**. A mutant allele that alters the survival rate between age $x$ and $x + 1$ changes inclusive fitness contributions between age classes according to the following:

$$dw_{yx} = \begin{cases} d\dot{p}(x) \text{ if } y = x + 1 \\ -d\dot{p}(x)\Big[\dot{h}(x) + \dot{k}(x)\hat{r}(x)\Big] \text{ if } y = 1 \\ 0 \text{ otherwise} \end{cases} \quad (7)$$

where $\dot{h}(x)$ is the proportion of offspring after dispersal at the local patch that are the direct and indirect contributions of a focal individual aged $x$, $\dot{k}(x)$ is the proportion of offspring that are born due to the genotypes of other individuals on the patch, and $\hat{r}(x)$ is the relatedness of an individual aged $x$ to the offspring of other patch mates (see 'Methods'). As we assume mortality occurs between breeding seasons, a focal individual's contributions to the survival and reproduction of other age classes are only affected at $x + 1$. Therefore, the effects of the allele are to increase individual survival (top row of (7)), and to reduce the probability a newborn offspring establishes on the patch through this increased individual survival (middle row of (7)).

Let $S_p(x)$ be the component of the force of selection due the effect of a mutant allele on the survival rate between age $x$ and $x + 1$. Using Eqs. (6) and (7), in a stationary population with limited dispersal and social interactions between individuals, this can be written as:

$$S_p(x) = \frac{d\dot{p}(x)}{d\delta} \frac{f_x(v_{x+1} - \Big[\dot{h}(x) + \dot{k}(x)\hat{r}(x)\Big]v_1)}{\mathbf{f} \cdot \mathbf{v}} \quad (8)$$

Equation (8) shows that the overall direction of the force of selection acting on a mutant allele that affects the survival rate between age $x$ and $x + 1$ is a balance of two forces: the inclusive reproductive value at age $x + 1$ *vs* the inclusive reproductive value of offspring (displaced by the survival of the focal individual) that have varying relatedness to the focal individual aged $x$. The term $\mathbf{f} \cdot \mathbf{v}$ acts to scale the forces of selection in terms of generation time[2,20].

**The inclusive fitness force of selection on reproduction**. A mutant allele that alters reproduction at age $x$ changes inclusive fitness contributions between age classes according to the following (see 'Methods' and Supplementary Information Appendix D):

$$dw_{yx} = \begin{cases} 0 \text{ if } y = x + 1 \\ d\dot{b}(x)\Big[(1-d)g(x)\Big[(1-\dot{h}(x)) - \dot{I}(x) - \dot{k}(x)\hat{r}(x)\Big] + (1-c)d\bar{g}\Big] \text{ if } y = 1 \\ 0 \text{ otherwise} \end{cases} \quad (9)$$

The effect of the mutant allele is only on the newborn age class. A small increase in $\dot{b}(x)$ will increase the likelihood an individual's offspring establishes on a patch. However, this effect is counter-acted at the individual's patch by the reduction in the probability of establishment for offspring born to other individuals on the patch (see 'Methods' and Supplementary Information Appendix D).

Then, let $S_m(x)$ be the component of the force of selection due the effect of a mutant allele on reproduction at age $x$. Using (6) and (9), in a stationary population with limited dispersal and

social interactions between individuals, this can be written as:

$$S_m(x) = \frac{d\dot{b}(x)}{d\delta}\frac{f_x v_1}{\mathbf{f}\cdot\mathbf{v}}\left[(1-d)g(x)\Big[(1-h(x))-\dot{I}(x)-\dot{k}(x)\hat{r}(x)\Big]+(1-c)d\bar{g}\right]$$

(10)

where

$$\dot{I}(x) = \frac{\sum_z T^x_{1,z}(1-d)}{b(x)(1-d)+(N-1)\bar{b}(1-d)+N\bar{b}(1-c)d}$$

(11)

is the fraction of all offspring at the local patch after dispersal that exist due to indirect effects of the genotype of a focal individual aged $x$. Equation (10) shows that the overall force of selection acting on a mutant allele that affects the rate of reproduction at age $x$ is also comprised of two components: (i) the effect of the allele on the probability of establishment of different types of offspring onto the local patch and (ii) the effect of the allele on the dispersing offspring that are part of the direct fitness of the focal individual aged $x$. Selection for effect (ii) will always be positive; however, selection for effect (i) will depend on the relative weights each class of offspring contributes to the overall effect. In this model, an increase in direct reproduction is, all else being equal, beneficial for the direct fitness of a focal individual, but can be detrimental to the indirect fitness of the focal individual, as it can reduce the probability of establishment of offspring born to related individuals.

**Applications of the model**. Equations (8) and (10) provide general solutions for age-specific inclusive fitness forces of selection on individual survival and reproduction in group structured populations. To visualise the results, we consider two hypothetical populations of iteroparous individuals with social interactions (Figs. 2 and 3). For each, we consider background demography described by age-specific vital rates, $p(x)$ and $b(x)$. We parameterise mortality risk at age $x$ using a modified version[14] of the Siler model[54]

$$\mu(x) = \alpha_1 e^{-\beta_1 x} + \alpha_2 e^{\beta_2 x}$$

(12)

This function allows mortality risk to decrease with age before sexual maturity (negative exponent in first term) and then increase again with age. The probability of survival at age $x$, $p(x)$, is then equal to $e^{-\mu(x)}$. The probability of survival to age $x$ is then $l(x) = \prod_1^{x-1}p(x)$, with $l(1) = 1$. As we assume all patches have no breeding positions available at the start of each breeding seasons (i.e., a stationary population), we can calculate the asymptotic frequency ($s_x$) of each age class as

$$s_x = \frac{l(x)}{\sum_y l(y)}$$

(13)

which is the right eigenvector associated with the dominant eigenvalue of the population projection matrix $\mathbf{A}$, before translating into inclusive fitness contributions. We then parameterise individual rate of reproduction at age $x$ as:

$$b(x) = \begin{cases} 0 & \text{if } x < \varepsilon \\ (x-\varepsilon)e^{-\varphi(x-\varepsilon)} & \text{if } x \geq \varepsilon \\ 0 & \text{if } x > \kappa \end{cases}$$

(14)

where $\varepsilon$ designates the age of reproductive maturity, $\kappa$ represents an age at which reproduction ceases, and $\varphi$ modulates the shape of reproduction across age classes.

Figures 2a and 3a illustrate the life cycles of the two hypothetical social populations. Figure 2a considers a population with post-reproductive individuals providing care for juveniles, as seen in humans[55], orcas[56], and Asian elephants[57]. Figure 3a considers a population with juvenile individuals providing help to the

reproduction adult breeders, as is found in many cooperatively-breeding species[31]. Figures 2b and 3b display the modelled survival, reproduction, and social transfers as a function of individual age. To allow comparison of our results with Hamilton's[2] forces of selection, we define $m(x)$ as the age-specific effective fecundity in a population with complete dispersal ($d = 1$), no transfers, and the same age structure as our population[14]:

$$m(x) = b(x)\frac{N(1-\bar{p})}{N\bar{b}}$$

(15)

Hamilton's forces of selection can then be computed for age-specific survival ($p(x)$) and fecundity ($m(x)$) in a stationary population of the same age-structure as our population (described by $\mathbf{A}$), where vital rates have not been translated into inclusive fitness contributions. The force of selection acting on survival is then $\frac{\sum_{y=x+1}^{\omega}l(y)m(y)}{T}$ and for reproduction is $\frac{l(x)}{T}$, with $T = \sum_{y=1}^{\omega}l(y)m(y)y$ giving the mean generation time[2,14]. Figures 2c and 3c then show the forces of selection acting on survival and reproduction at age $x$ using Hamilton's indicators, which can be compared to our forces of selection that take juvenile dispersal and transfers into account ((8) and (10)). Figure 2d and Fig. 3d display the age-specific relatedness of an individual to other individuals on their patch, given the specific parameters chosen (see 'Methods').

We show that the force of selection acting on survival in social populations is not necessarily constant before maturity, as predicted by classical theory[2]. The exact pattern depends on whether pre-reproductive individuals gain indirect fitness through transfers or not. When juveniles do not engage in helping behaviour, the force of selection tends to increase in the juvenile period as relatedness to newborn offspring decreases with increasing juvenile age (Fig. 2c, d; Supplementary Fig. 1; Supplementary Fig. 2). This decline in local relatedness facilitates a more 'selfish' force of selection on survival throughout the juvenile period. On the other hand, when juveniles provide help to adult reproduction, the force of selection on survival generally decreases from the age at which indirect fitness was first accrued (Fig. 3c; Supplementary Fig. 3), rather than the age of first reproduction. In both examples, the force of selection on survival then declines throughout adulthood as future inclusive reproductive value declines and the relatedness to newborn offspring increases. When post-reproductive adults continue to accrue indirect fitness, the force of selection on survival can remain above zero in post-reproductive age classes (Fig. 2c; Supplementary Fig. 1). The magnitude of the force of selection is greater in post-reproductive age classes when juvenile dispersal is lower (and so there is higher local relatedness) and the magnitude of help provided by post-reproductive individuals is higher (Supplementary Fig. 1). In general, the force of selection on survival will always have a positive component until the final age at which inclusive fitness is accrued, rather than necessarily the age of last reproduction. At this age, when future survival is no longer possible, the first term on the numerator of Eq. (8) is zero, and so, if there is some level of local relatedness (i.e. $\hat{r}(x)>0$), selection will favour increased mortality as it will benefit the establishment of related juveniles.

Incorporating potential harm to relative's reproductive output through increased personal reproduction changes the trajectory of the force of selection acting on reproduction across age classes, and generally means a weaker force of selection on increasing reproduction compared to Hamilton's predictions (Figs. 2c and 3c). Inclusive fitness forces of selection acting on reproduction at age $x$ generally decline from birth, as is predicted by Hamilton's model[2], but not always (Supplementary Fig. 4), and

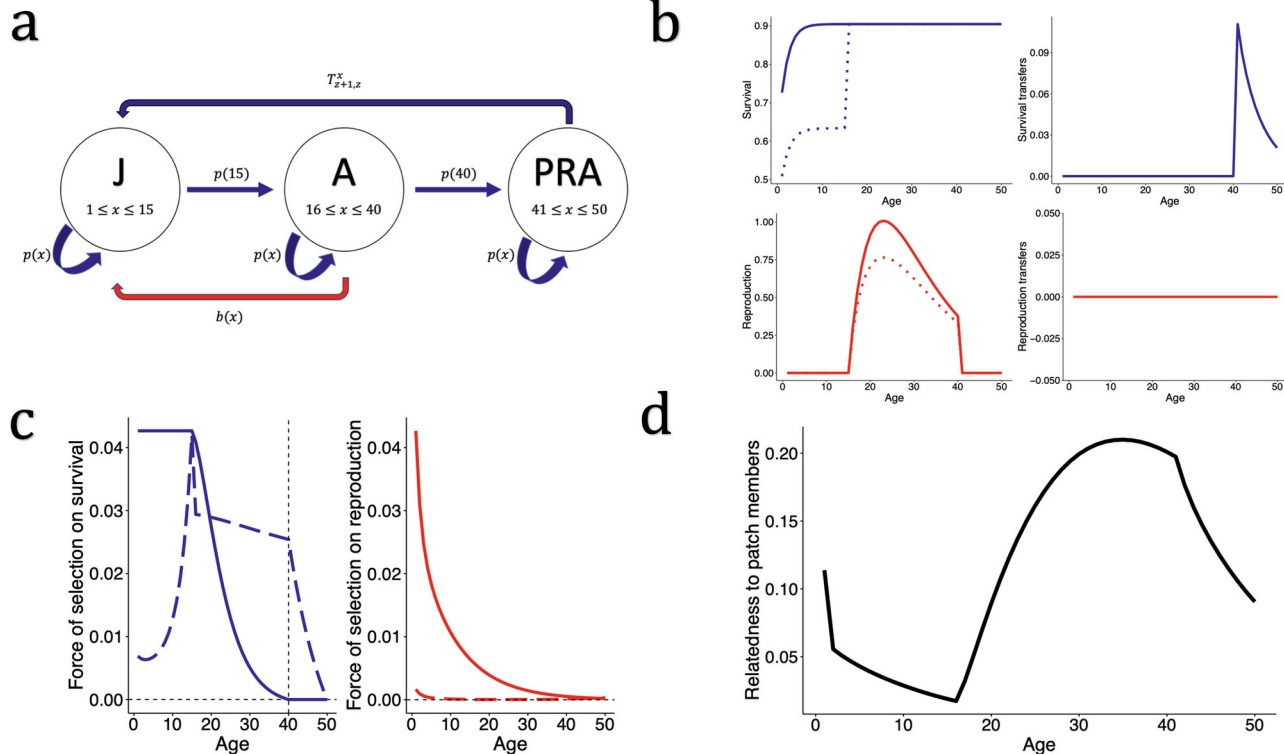

**Fig. 2 Age specific forces of selection in a social population with post-reproductive help. a** A hypothetical population of iteroparous individuals classified into three life cycle stages: juvenile (J), reproductive adult (A), and post-reproductive adult (PRA). The red arrow from A to J represents the reproduction of adult individuals, whereas the dark blue arrow from PRA to J represents the social contributions from post-reproductive adults to the survival of juveniles. **b** The specified rates of survival (top left) displaying $p(x)$ (continuous line) and $\dot{p}(x)$ (dotted line), reproduction (bottom left) displaying $m(x)$ ((15), continuous line) and $\dot{F}(x)D$ ((4–5), dotted line), and transfers to survival (top right) and reproduction (bottom right). The survival probability at age $x$ ($p(x)$) is produced from a Siler model (12) with parameters: $\alpha_1 = 0.4$, $\beta_1 = 0.6$, $\alpha_2 = 0.1$, $\beta_2 = 0$. Survival stripped of social effects ($\dot{p}(x)$) discounts the survival attributed to help from other age classes (2). Reproduction at age $x$ ($b(x)$) is modelled according to (14) with parameters: $\varepsilon = 15$, $\varphi = 0.125$, and $\kappa = 40$, and $m(x)$ displays the effective fecundity with full dispersal ($d = 1$). $\dot{F}(x)D$ represents the contribution to offspring through both direct and indirect reproduction, considering dispersal. In this model $d = 0.5$, $N = 4$, and $c = 0$. In this model, post-reproductive age classes contribute to the survival of younger individuals, whereas there are no social transfers in reproduction. **c** Hamilton's forces of selection (continuous lines) *vs.* inclusive fitness forces of selection (dashed lines). The inclusive fitness force of selection acting on survival at age $x$ increases during the juvenile period and then decreases but remains above zero in the post-reproductive period. The inclusive fitness force of selection acting on reproduction at age $x$ is weaker than the force of selection acting on survival and declines from birth until increasing in late adulthood when relatedness to other individuals declines. **d** The relatedness of an individual aged $x$ to another random individual on the patch declines throughout the juvenile (pre-reproductive) window, and then increases during adult reproduction before declining again as reproduction ceases. See Supplementary Figs. 1–4 for sensitivity analyses on model parameters.

the decline is more rapid when the rate of dispersal is lower (Supplementary Fig. 2). This more rapid decline is likely due to the greater inclusive fitness costs of increasing personal reproduction when local relatedness is higher. The force of selection on reproduction in early life is also weaker when post-reproductive adults have a more significant impact on juvenile survival. In all iterations of the model (Fig. 3c; Supplementary Fig. 3), there was a slight increase in the force of selection acting on reproduction in the final age class, when the force of selection on rate of survival becomes negative.

## Discussion
When considering the evolution of demographic senescence, evolutionary biologists use population growth rate, $r$, as the measure of fitness[38] (but see ref. [58]). The magnitude of the change in population growth rate due to an age-specific change in survival and/or reproduction generally declines with age (but see ref. [59] for other indicators of the force of selection), and this decline facilitates the evolution of senescence[2]. However, for social species, it is crucial to consider the inclusive fitness of individuals as the quantity that natural selection seeks

to maximise[7,8,10]. The change in inclusive fitness due to an age-specific change in individual survival and/or reproduction then considers the combined effect on all individuals that are affected by the change[39]. Here, we show that considering the inclusive fitness effects of a 'senescent' allele significantly alters the forces of selection acting on age-specific survival and reproduction.

Our framework provides several key insights into the force of selection acting on survival and reproduction in social species. First, the force of selection acting on the survival rate of that age class is the product of future inclusive reproductive value (IRV), rather than conventional reproduction value (RV[38]), and a modified version of the asymptotic frequency (stationary age distribution) of that age class that accounts for social transfers. Since IRV remains above zero after reproduction ceases, if post-reproductive adults continue to accrue indirect fitness benefits, then selection on survival of post-reproductive age classes does not necessarily go to zero as in Hamilton's model[2]. Importantly, this finding provides a formal inclusive fitness framework to the 'grandmother hypothesis'[60,61], supporting work that has suggested indirect fitness benefits are a key driver of

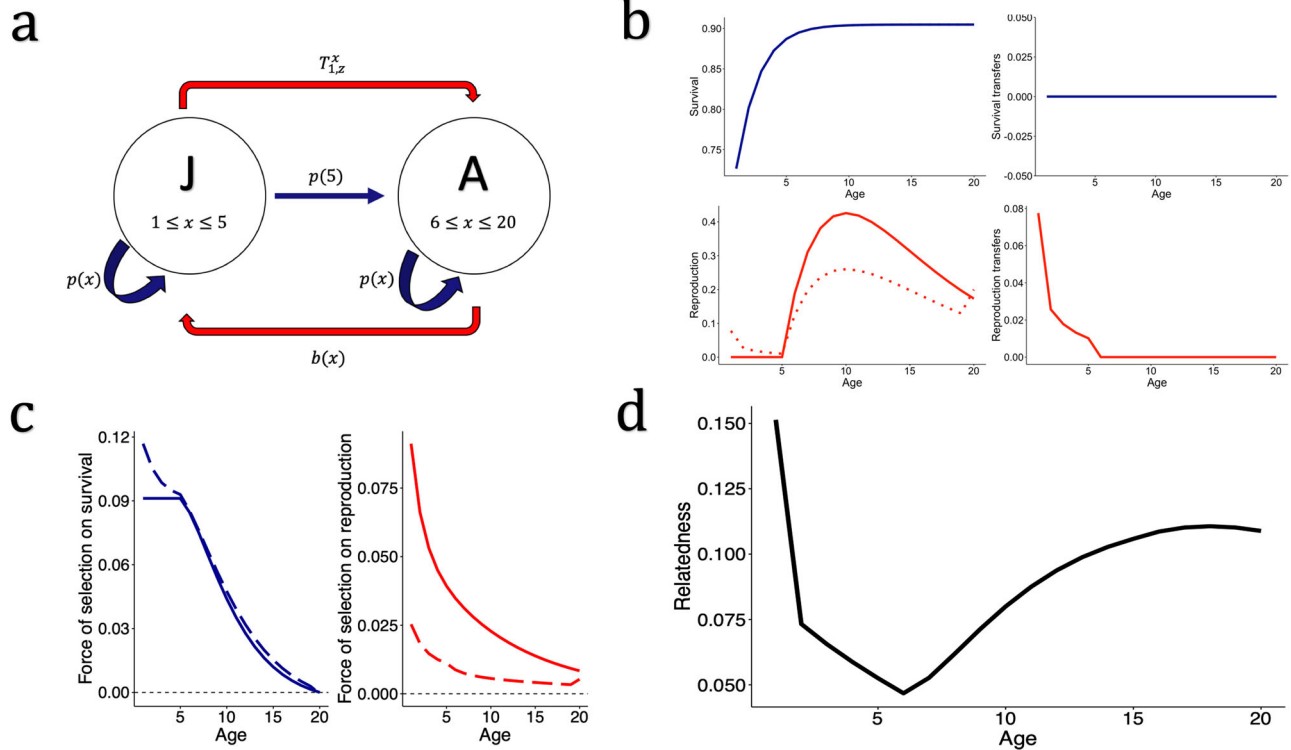

**Fig. 3 Age specific forces of selection in a social population with pre-reproductive help. a** A hypothetical population of iteroparous individuals with two life cycle stages: juvenile (J) and reproductive adult (A). The red arrow from A to J represents the reproduction of adult individuals, whereas the red arrow from J to A represents the social contributions from juveniles to the reproduction of adults. Note that, here, help is in the currency of reproduction, rather than survival (see Fig. 2a). **b** The specified rates of survival (top left) displaying $p(x)$ (continuous line), reproduction (bottom left) displaying $m(x)$ ((15), continuous line) and $\dot{F}(x)D$ ((4–5), dotted line), and transfers to survival (top right) and reproduction (bottom right). The survival probability at age $x$ ($p(x)$) is produced from a Siler model (12) with parameters: $\alpha_1 = 0.4$, $\beta_1 = 0.6$, $\alpha_2 = 0.1$, $\beta_2 = 0$. Note, unlike Fig. 2b, because there are no transfers of survival, $p(x) = \dot{p}(x)$. The rate of reproduction at age $x$ is modelled according to (14) with parameters: $\varepsilon = 5$, $\varphi = 0.2$, and $\kappa = 21$, and $m(x)$ displays the effective fecundity with full dispersal ($d = 1$). $\dot{F}(x)D$ represents the contribution to offspring through both direct and indirect reproduction, considering dispersal. In this model $d = 0.5$, $N = 4$, and $c = 0$. In this model, pre-reproductive age classes contribute to the reproduction of adult individuals, whereas there are no social transfers in reproduction. **c** Hamilton's forces of selection (continuous lines) vs. inclusive fitness forces of selection (dashed lines). The inclusive fitness force of selection acting on survival at age $x$ declines from birth. The inclusive fitness force of selection acting on reproduction at age $x$ is weaker than the force of selection on survival and also declines from birth but then increases in the final age class. **d** The relatedness of an individual aged $x$ to another random individual on the patch declines throughout the juvenile period, and then increases during adult reproduction.

post-reproductive lifespan[17,18,26]. In our framework, the force of selection on survival will remain above zero until an individual's future IRV is outweighed by the negative effects of its continued survival on the establishment of related offspring. Although, in our model, a negative force of selection on survival may be an artifact of enforcing a final age class, in theory, a kin-selected terminal investment strategy, in which it pays to invest heavily in reproduction at the expense of survival to maximise the establishment of kin, could be favoured[24].

The incorporation of age-specific indirect fitness into the evolutionary theory of senescence means that selection on survival before maturity is not necessarily constant (Figs. 2c and 3c). This difference occurs because of the balance between the future IRV of the individual and the IRV of newborns displaced by increased survival. If relatedness to other individuals declines throughout the juvenile period as a focal individual ages, and the focal individual's own IRV increases as they approach maturity, the balance in Eq. (8) is weighted more heavily towards the first term, and the force of selection on increased survival will increase. On the other hand, in populations where juveniles help and accrue indirect fitness, the force of selection on survival will decline from the age at which indirect fitness is first gained. This result implies that, in species with pre-reproductive help, senescence should start from the age at which inclusive fitness is first

gained, rather than the age of first reproduction, as in conventional models[2,14].

An inclusive fitness force of selection acting on reproduction depends on the costs and benefits associated with increasing personal reproduction at a given age. In our framework, selection for increased reproduction will always have a positive component due to the increased probability of an offspring (whether philopatric or dispersive) establishing on to a patch. However, the subsequent decrease in probability of other locally produced offspring establishing on to the patch reduces the magnitude of the force of selection acting on reproduction. This result may be especially important for groups experiencing strong competition over resources[12]. For example, a negligible force of selection on reproduction may favour reproductive restraint by some individuals within cooperatively-breeding groups, when access to reproduction is limited and inclusive fitness costs of increasing personal reproduction would be substantial[27].

Our model makes several simplifying assumptions. We assumed time-invariant vital rates ($p(x)$ and $b(x)$) and transfers between individuals, and a stationary population enforced by density-dependent juvenile establishment onto patches. Using a patch model with distinct individuals allows us to consider explicit interactions between individuals, while enforcing some form of kin competition through density-dependence (via

maximum patch size) allows us to examine how age-dependent inclusive fitness shapes the forces of selection. However, we chose to model a stationary population (all sites on a patch are filled at the start of a breeding seasons) to avoid the complication of empty sites on a patch. Hamilton's[2] forces of selection were computed for a population growing at any stable rate (positive, negative, or stationary), and allowing for this less restrictive condition may highlight the extent to which our results are due, in part, to our stationary population condition. In addition, at small patch population sizes (i.e. small $N$) demographic stochasticity is likely to be an important factor. Here, we made the restrictive assumption that all individuals except the mutant survive and reproduce at population average rates. In reality, demographic stochasticity is likely to have important consequences on population structure and relatedness, impacting inclusive fitness forces of selection, especially at a small value of $N$. An extension to our model which incorporates demographic stochasticity (perhaps following ref. [21]) is a potential further avenue of future study that could expand the generality of our results. A further important assumption we made was that some set of social interactions were stable in the population (just as vital rates are assumed to be stable in most evolutionary models). Our methodology then allows derivations of inclusive fitness forces of selection from these social interactions (transfers). Although the two examples we provided are general representations of social systems in nature, expanding our approach to modelling the co-evolution of social and 'senescent alleles' may provide more robust predictions when social traits have evolved to some stable equilibrium, or are continually evolving. Additionally, previous work has suggested that empty sites on patches can be influential in determining the spread of cooperative alleles[62]. An extension to the model which considers both stable (not stationary) population growth and co-evolution will likely be a highly technical but fruitful avenue for future work. Finally, our approach was, for the most part, unconcerned with genetics and correlations between traits[63–65]. Genetic variance and covariance among traits and mutation rates also influence the evolution of traits and would be an important future step for further developments of the model[65].

All this notwithstanding, the model we present here provides a framework from which to expand our understanding of senescence across social species. For example, previous work has found mixed evidence for extended lifespan in cooperative breeders[66–68], and some evidence for differences in rates of senescence between cooperative and non-cooperative breeders[69]. Previous theory suggests that it is longer life and overlapping generations that initially favour cooperation[32], but also that a delayed age of first reproduction as a result of queuing for reproduction might be a self-reinforcing mechanism for extended lifespan in cooperative breeders[70]. However, multiple other facets of the demography of cooperative breeding systems, including the process of group formation[71], differences in phenotype between subordinates and dominants resulting in differential age-specific mortality and reproduction[72], and the structure of dominance hierarchies[73] all have the potential to play a role in determining lifespan and rates of senescence. These group-level characteristic can, in theory, contribute to the shape of the age class asymptotic frequency and inclusive reproductive value distributions that, demonstrated in this model, underpin inclusive fitness forces of selection. Our model provides a framework to stimulate further theoretical work for how these features of cooperative breeding systems may impact the evolution of lifespan and senescence.

Here, we focused on how cooperative interactions between members of a group can alter age-specific inclusive fitness forces of selection. However, in many groups, competitive interactions over limited resources are also rife. In our model, transfers between age classes reflect the net effect of the presence of an individual in one age class on the survival and reproduction of an individual in another age class. If the net effect is negative, then the genetic offspring transfer is also negative. For example, consider again the social system illustrated in Fig. 2. Instead of post-reproductive individuals having a positive effect of the survival of juveniles, let us instead imagine a scenario in which the presence of post-reproductive individuals is harmful to the survival of juveniles. An allele that increases the rate of survival in such post-reproductive individuals will be selected against due to the inclusive fitness costs imposed from the negative effects on related juvenile individuals, potentially hastening the evolution of more rapid senescence. Finally, we only considered indirect fitness returns from social interactions. In many cooperative breeding systems, however, direct fitness returns from social interactions can also explain alloparental care[31]. In such systems, an individual's social transfers would ultimately benefit its own future survival or reproduction, (as well as its indirect fitness, where recipients of help are kin), as hypothesised by group augmentation theory[74].

In summary, recent research has focused on the potential for social interactions to drive variation in lifespan and rates of senescence across species[1,75]. In an attempt to understand the mechanisms for how this might occur, we derived the inclusive fitness effects of a senescent allele in hypothetical group structured populations composed of kin and non-kin that interact with one another. When inclusive fitness consequences of individual survival or reproduction are considered, age-specific forces of selection can vary markedly from previous asocial models. Further theoretical, empirical and comparative studies are now needed to determine the amount of variation in lifespan and rates of senescence that can be explained by social modes of life.

## Methods

**Appendix A: Relatedness**. In order to quantify indirect genetic contributions, it is essential to consider the relatedness between different age classes of individuals in the population. We can describe the relatedness of a focal individual aged $x$ to other individuals on the patch, including themselves, as:

$$r(x) = \frac{1}{N} + \frac{N-1}{N}\hat{r}(x) \tag{A1}$$

Then, let $r_{yx}$ denote the probability that an allele sampled randomly from a given locus in an individual aged $x$ is identical by descent (IBD) to an allele sampled randomly from the same locus in an individual aged $y$[35,76,77]. The term $\hat{r}(x)$ represents the average relatedness of a breeding individual aged $x$ to another random breeder on the same patch[26,27], which is equivalent to the mean relatedness of a focal individual aged $x$ across all age classes ($\hat{r}(x) = \bar{r_{yx}}$). Given the assumption of haploid genetics and asexuality, $\hat{r}(x)$ is therefore also the relatedness of a focal individual aged $x$ to the offspring of the other individuals on the patch. Under the assumption of infinite patches, any immigrants arriving at the focal patch will not have any relatives when they arrive, and the relatedness of individuals on the patch of any age to these immigrants is equal to 0.

Let us define $h(x)$ as the proportion of offspring after dispersal at the local patch that are the offspring (not partitioned into inclusive fitness contributions) of a focal individual aged $x$:

$$h(x) = \frac{b(x)(1-d)}{b(x)(1-d) + (N-1)\bar{b}(1-d) + N\bar{b}d(1-c)} \tag{A2}$$

where $\bar{b}$ represents the average rate of reproduction. For simplicity, we assume no demographic stochasticity within or between patches (see 'Discussion'). Then, let $k(x)$ define the

proportion of offspring after dispersal at the local patch that are the offspring (not partitioned into inclusive fitness contributions) of other individuals on the patch besides the focal individual aged $x$:

$$k(x) = \frac{(N-1)\bar{b}(1-d)}{b(x)(1-d) + (N-1)\bar{b}(1-d) + N\bar{b}d(1-c)} \quad (A3)$$

Using Eqs. (A2) and (A3), we can describe the relatedness between an individual aged $x$ to a different individual on the patch aged $y$ as a function of both individual's ages:

$$r_{yx} = \begin{cases} h(x-y) + k(x-y)\hat{r}(x-y) & \text{if } y < x \\ \frac{(1-d)^2}{(1-cd)^2}\left[\bar{h}^2 + \left(1-\bar{h}^2\right)\hat{r}(1)\right] & \text{if } y = x \\ h(y-x) + k(y-x)\hat{r}(1) & \text{if } y > x \end{cases} \quad (A4)$$

First, consider the case when the individual of age $x$ is older than the individual of age $y$ (top row of (A4)). The individual aged $y$ was born $y$ breeding seasons ago, when the individual aged $x$ was $x - y$ years old. At age $x - y$, the proportion of offspring at the local patch after dispersal that are the offspring of an individual aged $x - y$ is defined as $h(x-y)$. Therefore, with probability $h(x-y)$, the individual aged $y$ is the offspring of the individual aged $x$ from $x - y$ breeding seasons ago, and thus the relatedness between the two individuals is one. Then, let $k(x-y)$ define the proportion of offspring at the local patch after dispersal $x - y$ breeding seasons ago that were the offspring of other individuals on the patch besides the individual now aged $x$. With probability $k(x-y)$, the individual aged $y$ was born to another individual on the patch besides the individual now aged $x$. Therefore, the relatedness of the individual aged $x$ to the individual aged $y$ is equal to the relatedness of an individual aged $x - y$ to a random offspring born locally to the patch, which is equal to the relatedness of an individual aged $x - y$ to another random individual on the patch ($\hat{r}(x-y)$). The remaining proportion of offspring at the patch after dispersal $x - y$ breeding seasons ago $(1 - h(x-y) - k(x-y))$ were from elsewhere in the population and thus relatedness is 0.

Second, consider the case when both individuals are the same age (second row of (A4)). The probability that both are local to the patch is $\frac{(1-d)^2}{(1-cd)^2}$. If both individuals are born locally, we then have to consider the probability that both individuals were born to the same mother, and thus are siblings related by 1. We can first define the average proportion across age classes of offspring competing for a site on the patch that are born to an individual that reproduces as a function of its age, when all other individuals reproduce at the population average level, as $\bar{h}$. Then, the probability that two offspring born $x$ breeding seasons ago were born to the same mother is equal to $\bar{h}^2$. One minus $\bar{h}^2$ is then the probability that these two locally born offspring $x$ breeding seasons ago were born to different mothers, in which case the relatedness of an individual aged $x$ to a same aged individual is equal to the relatedness of an individual to a random member of the patch at age 1 when the focal individual established onto the patch ($\hat{r}(1)$). The final scenario (bottom row of (A4)) considers the case when the individual aged $y$ is older than the individual aged $x$. In this case the logic is similar to the case when the individual aged $x$ is older than the individual aged $y$. With probability $h(y-x)$, the individual aged $x$ is the offspring of an individual aged $y$, and so relatedness is 1. With probability $k(y-x)$, the individual aged $x$ is the offspring of another individual on the patch at time $y - x$, when the individual aged $x$

was 1. Therefore, the relatedness to the individual aged $y$ is equal to the average relatedness of a newborn that doesn't disperse upon establishing ($\hat{r}(1)$). With probability $(1 - h(y-x) - k(y-x))$ the individual aged $x$ dispersed from elsewhere in the population and so relatedness is equal to 0.

To calculate $\hat{r}(x)$, the average relatedness of an individual aged $x$ to another individual on the patch, we need to calculate the average relatedness of individuals aged $x$ to all other age classes. Using each possible relatedness between age classes (A4), we can do this by weighting each age class specific relatedness term by the asymptotic frequencies ($s_x$) of the relevant age classes:

$$\hat{r}(x) = \left(\sum^{y<x} s_y\left[(h(x-y) + k(x-y)\hat{r}(x-y))\right]\right)$$
$$+ s_x\frac{(1-d)^2}{(1-cd)^2}\left(\bar{h}^2 + \left(1-\bar{h}^2\right)\hat{r}(1)\right) \quad (A5)$$
$$+ \left(\sum_{y=x+1}^{y=\omega} s_y\left[h(y-x) + k(y-x)\hat{r}(1)\right]\right)$$

*Deriving.* $\hat{r}(1)$

To find a general solution for $\hat{r}(1)$, which is the relatedness of an individual aged 1 to another random breeder on the patch, let us consider a case of a population with 3 age classes ($\omega = 3$). Using the logic that $x = 1$ is the first age class and therefore $y$ cannot be younger than $x$, $\hat{r}(1)$ with 3 age classes becomes:

$$\hat{r}(1) = s_1\frac{(1-d)^2}{(1-cd)^2}\left[\bar{h}^2 + \hat{r}(1)\left(1-\bar{h}^2\right)\right] + \sum_{y=2}^{3} s_y\left[h(y-x) + k(y-x)\right]\hat{r}(1) \quad (A6)$$

Expanding the summation term:

$$\hat{r}(1) = s_1\frac{(1-d)^2}{(1-cd)^2}\left[\bar{h}^2 + \hat{r}(1)\left(1-\bar{h}^2\right)\right] + s_2[h(1) + k(1)\hat{r}(1)] + s_3[h(2) + k(2)\hat{r}(1)] \quad (A7)$$

Expanding out each term:

$$\hat{r}(1) = s_1\frac{(1-d)^2}{(1-cd)^2}\bar{h}^2 + s_1\frac{(1-d)^2}{(1-cd)^2}\hat{r}(1)\left(1-\bar{h}^2\right) + s_2h(1) + s_2k(1)\hat{r}(1) + s_3h(2) + s_3k(2)\hat{r}(1) \quad (A8)$$

Factoring on the RHS by $\hat{r}(1)$:

$$\hat{r}(1) = \hat{r}(1)\left[s_1\frac{(1-d)^2}{(1-cd)^2}\left(1-\bar{h}^2\right) + s_2k(1) + s_3k(2)\right]$$
$$+ s_1\frac{(1-d)^2}{(1-cd)^2}\bar{h}^2 + s_2h(1) + s_3h(2) \quad (A9)$$

Re-arranging, and factoring on the LHS by $\hat{r}(1)$:

$$\hat{r}(1)\left[1 - \left[s_1\frac{(1-d)^2}{(1-cd)^2}\left(1-\bar{h}^2\right) + s_2k(1) + s_3k(2)\right]\right] = s_1\frac{(1-d)^2}{(1-cd)^2}\bar{h}^2 + s_2h(1) + s_3h(2) \quad (A10)$$

Dividing both sides by $[1 - [s_1\frac{(1-d)^2}{(1-cd)^2}(1-\bar{h}^2) + s_2k(1) + s_3k(2)]]$:

$$\hat{r}(1) = \frac{s_1\frac{(1-d)^2}{(1-cd)^2}\bar{h}^2 + s_2h(1) + s_3h(2)}{1 - \left[s_1\frac{(1-d)^2}{(1-cd)^2}\left(1-\bar{h}^2\right) + s_2k(1) + s_3k(2)\right]} \quad (A11)$$

Finally, to generalise for all possible number of age classes, we can re-write (A11) as

$$\hat{r}(1) = \frac{s_1\frac{(1-d)^2}{(1-cd)^2}\bar{h}^2 + \sum_{y=2}^{\omega}s_yh(y-1)}{1 - \left[s_1\frac{(1-d)^2}{(1-cd)^2}\left(1-\bar{h}^2\right) + \sum_{y=2}^{\omega}s_yk(y-1)\right]} \quad (A12)$$

Once we have $\hat{r}(1)$, $\hat{r}(x)$ for all other age classes can be solved recursively.

## Appendix B: Analytical solutions

*The effect of a mutant allele that alters age-specific survival in a social population.* Let us first consider how, in a resident population with limited dispersal and social interactions, a mutant allele that affects survival at age $x$ will alter the number of class-$y$ offspring of a focal individual aged $x$. First, the most obvious effect of this allele is to change the individual's probability of survival to the next breeding season, which is $d\dot{p}(x)$. A change in survival will also alter the contributions a focal individual aged $x$ makes to the offspring class, $w_{1x}$. For example, if the mutant allele increases survival at age $x$, then there is a greater chance the focal individual survives to age $x + 1$, and this subsequently reduces the probability that an offspring at the focal patch after dispersal will establish onto the patch before the next breeding season. Four classes of offspring will exist at the focal patch after dispersal: (1) the offspring of a focal individual aged $x$, (2) the offspring of other individuals on the patch that exist due to the genotype of a focal individual aged $x$, (3) the offspring of other individuals on the patch that don't owe their existence to the genotype of a focal individual aged $x$, and (4) offspring from elsewhere in the population. As we are interested in the inclusive fitness effect of the mutant allele, we must consider the fates of all the offspring that are impacted by the effect of the allele[24].

We can consider the first two sets of offspring together and ask how a change in survival at age $x$ alters the direct and indirect production of offspring of a focal age $x$ individual (working showed below).

$$\frac{dw_{1x}(1,2)}{d\dot{p}(x)} = \dot{F}(x)\big[(1-d)g(x) + (1-c)d\bar{g}\big] - \dot{F}(x)\big[(1-d)g'(x) + (1-c)d\bar{g}\big]$$

(B1)

with $g'(x)$ displaying that the effect of the allele is to alter the probability that the direct and indirect offspring of the individual aged $x$ establish on to the patch. Equation (B1) can be worked through and simplified as:

$$\frac{dw_{1x}(1,2)}{d\dot{p}(x)} = \dot{F}(x)(1-d)g(x) + \dot{F}(x)(1-c)d\bar{g} - \dot{F}(x)(1-d)g'(x) - \dot{F}(x)(1-c)d\bar{g}$$

$$= \dot{F}(x)(1-d)g(x) - \dot{F}(x)(1-d)g'(x)$$

$$= \dot{F}(x)(1-d)[g(x) - g'(x)]$$

$$= \dot{F}(x)(1-d)\left[\frac{1 - p(x) + (N-1)(1-\bar{p})}{b(x)(1-d) + (N-1)\bar{b}(1-d) + N\bar{b}(1-c)d}\right.$$
$$\left. - \frac{1 - p'(x) + (N-1)(1-\bar{p})}{b(x)(1-d) + (N-1)\bar{b}(1-d) + N\bar{b}(1-c)d}\right]$$

$$= \dot{F}(x)(1-d)\left[\frac{-d\dot{p}(x)}{b(x)(1-d) + (N-1)\bar{b}(1-d) + N\bar{b}(1-c)d}\right]$$

$$= -d\dot{p}(x)\left[\frac{\dot{F}(x)(1-d)}{b(x)(1-d) + (N-1)\bar{b}(1-d) + N\bar{b}(1-c)d}\right]$$

Finally, let $\dot{h}(x) = \frac{\dot{F}(x)(1-d)}{b(x)(1-d) + (N-1)\bar{b}(1-d) + N\bar{b}(1-c)d}$ be defined as the proportion of offspring at the focal patch after dispersal that are born due the genotype of a focal individual aged $x$. Note, $\dot{h}(x)$ is different from $h(x)$ (see 'Methods' Appendix A), as $h(x)$ does not partition the offspring with respect to inclusive fitness contributions. The relatedness of the indirect offspring has already been discounted in the calculation of $\dot{F}(x)$, and the relatedness of a focal individual to its own offspring is 1, so we can re-write (B1) as

$$\frac{dw_{1x}(1,2)}{dp(x)} = -d\dot{p}(x)\dot{h}(x)$$

(B2)

Let us now consider the third set of offspring and ask how a change in survival of a focal individual at age $x$ impacts the offspring of other individuals on the patch that don't owe their existence to the genotype of a focal individual aged $x$. In the resident population, this contribution is 0. However, an increase in survival of an individual aged $x$, for example, will reduce the

likelihood that any of these offspring that do not disperse will establish onto the patch before the next breeding season. We can write the average number of offspring of all individuals on the patch, in the presence of a focal individual aged $x$, that will establish onto the local patch as

$$(N-1)\bar{F}(1-d)g(x)$$

(B3)

The effect of a mutant allele that alters the survival of a focal individual aged $x$ on this expected number of offspring can then be written as

$$\frac{dw_{1x}(3)}{d\dot{p}(x)} = (N-1)\bar{F}(1-d)g(x) - (N-1)\bar{F}(1-d)g'(x)$$

(B4)

Equation (B4) can then be worked through and simplified as

$$\frac{dw_{1x}(3)}{d\dot{p}(x)} = (N-1)\bar{F}(1-d)[g(x) - g'(x)]$$

$$= (N-1)\bar{F}(1-d)\left[\frac{1 - p(x) + (N-1)(1-\bar{p})}{b(x)(1-d) + (N-1)\bar{b}(1-d) + N\bar{b}(1-c)d}\right.$$
$$\left. - \frac{1 - p'(x) + (N-1)(1-\bar{p})}{b(x)(1-d) + (N-1)\bar{b}(1-d) + N\bar{b}(1-c)d}\right]$$

$$= (N-1)\bar{F}(1-d)\left[\frac{-d\dot{p}(x)}{b(x)(1-d) + (N-1)\bar{b}(1-d) + N\bar{b}(1-c)d}\right]$$

$$= -d\dot{p}(x)\left[\frac{(N-1)\bar{F}(1-d)}{b(x)(1-d) + (N-1)\bar{b}(1-d) + N\bar{b}(1-c)d}\right]$$

Similar to the logic above, let $\dot{k}(x) = \frac{(N-1)\bar{F}(1-d)}{b(x)(1-d) + (N-1)\bar{b}(1-d) + N\bar{b}(1-c)d}$ be defined as the proportion of offspring at the focal patch after dispersal that are average direct and indirect offspring of all other individuals bar the focal individual aged $x$. These offspring are related to the focal individual by $\hat{r}(x)$ and so the above becomes

$$\frac{dw_{1x}(3)}{d\dot{p}(x)} = -d\dot{p}(x)\dot{k}(x)\hat{r}(x)$$

(B5)

Given our assumptions of an infinite population, we can assume that relatedness of any individual on a patch to offspring that have dispersed from elsewhere will be equal to zero. Therefore, the relatedness of a focal individual aged $x$ to the proportion of offspring after dispersal that were not born locally on the patch is zero. Thus, there is an overall balance of the effect of the mutant allele on a focal individual of age $x$'s production of newborns weighted on one side by locally produced offspring (with varying relatedness) and on the other side by dispersed offspring. The total effect of a mutant allele that alters age-specific survival on the production of offspring can then be summed as

$$\frac{dw_{1x}}{d\dot{p}(x)} = -d\dot{p}(x)\dot{h}(x) - d\dot{p}(x)\dot{k}(x)\hat{r}(x) = -d\dot{p}(x)[\dot{h}(x) + \dot{k}(x)\hat{r}(x)]$$

(B6)

The overall effect ($dw_{yx}$ for all y) of a mutant allele that alters age-specific survival is then shown in (6) in the main text.

*The effect of a mutant allele that alters age-specific reproduction in a social population.* Let us now consider how a mutant allele that affects reproduction at age $x$ will alter the class-$y$ offspring a focal individual aged $x$ in our social population. First, we assume for simplicity that a change in reproduction of a focal individual aged $x$ does not alter the individual's probability of survival to the next breeding season, or its contributions to the survival of other individuals alive on the patch. These are obvious extensions for future iterations of the model (see 'Discussion'). We therefore limit the effects of a change in reproduction to altering the contributions a focal individual aged $x$ makes to the offspring class, $w_{1x}$. There are four different types of offspring to consider: (1) the offspring of a focal individual aged $x$ that exist due to its own genotype, (2) the offspring of other individuals on the patch that exist due to the genotype of a focal individual aged $x$, (3) the

offspring of other individuals on the patch that don't owe their existence to the genotype of a focal individual aged $x$, and (4) offspring from elsewhere in the population. Again, as we are interested in the inclusive fitness effect of the mutant allele, we must consider the fates of all the offspring that are impacted by the effect of the allele[24].

The inclusive fitness effects of a mutant allele that causes a change in the direct rate of reproduction of a focal individual aged $x$ for each class of offspring can be displayed as follows:

$$\frac{dw_{1x}(1)}{d\dot{b}(x)} = \dot{b}(x)\left[(1-d)g(x) + (1-c)d\bar{g}\right] - \dot{b}\prime(x)\left[(1-d)g\prime(x) + (1-c)d\bar{g}\right]$$
(B7)

$$\frac{dw_{1x}(2)}{d\dot{b}(x)} = \sum_z T^x_{1,z}(1-d)g(x) - \sum_z T^x_{1,z}(1-d)g\prime(x)$$
(B8)

$$\frac{dw_{1x}(3)}{d\dot{b}(x)} = (N-1)\bar{F}(1-d)g(x) - (N-1)\bar{F}(1-d)g\prime(x)$$
(B9)

with prime notation displaying that the explicit effects of the allele. Above, (B7) considers the effect of the allele on the focal individual's direct production of offspring, (B8) the effect of the allele on the indirect offspring of focal, and (B9) the effect on offspring born to other individuals on the patch not due to the genotype of focal, but whom focal might be related to more than the population average (zero). Again, individuals that disperse from elsewhere in the population to the focal patch are assumed to be related to any individual on the patch by zero, and so the inclusive fitness effect of the allele with regards to the fourth class of offspring is also equal to zero. Furthermore, given our assumption of infinite patches, the effect of the allele on the second and third classes of offspring is limited to those offspring which do not disperse i.e. compete for a site at the local patch. The simplification of (B7)–(B9) follows the same logic as (B1)–(B5). The resulting derivations are lengthy and so are available in the Supplementary Information (Appendix D). The overall effect of the mutant allele that causes a change in the rate of reproduction of a focal individual aged $x$ is the sum of the effects (B7)–(B9) and can be expressed as:

$$\frac{dw_{1x}}{d\dot{b}(x)} = d\dot{b}(x)\left[(1-d)g(x)\left[(1-h(x)) - \dot{I}(x) - \dot{k}(x)\hat{r}(x)\right] + (1-c)d\bar{g}\right]$$
(B10)

The overall effect ($dw_{yx}$ for all y) of a mutant allele that alters age-specific reproduction is then shown in (8) in the main text.

**Reporting summary**. Further information on research design is available in the Nature Portfolio Reporting Summary linked to this article.

## Code availability

The R code to produce Figs. 2 and 3 is available as Supplementary R Code.

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

## Acknowledgements

We thank H. Caswell, A. Leeks, J. Park, M. Patel and T. Scott for helpful discussions and feedback. We also thank four anonymous reviewers for helpful critique that improved the manuscript. M.R. was supported by funding from the Biotechnology and Biological Sciences Research Council (BBSRC) (BB/M011224/1). R.S.-G. was supported by a NERC Independent Research Fellowship (NE/M018458/1). This research was funded in whole, or in part, by the UKRI [BB/M011224/1; NE/M018458/1]. For the purpose of Open Access, the authors have applied a CC BY public copyright licence to any Author Accepted Manuscript version arising from this submission.

## Author contributions

M.R. and M.B.B. conceived the idea for the model. J.P.G., R.S.-G., M.R. and M.B.B. contributed to the development of the model, and applications. M.R. drafted the first version and together with J.P.G., R.S.-G. and M.B.B. revised and edited the manuscript.

## Competing interests

The authors declare no competing interests.
