## [Peer Review File · Communications Biology]

Reviewers' comments:

Reviewer #1 (Remarks to the Author):

General feedback:

Thank you for this nice manuscript, which I found to be very well written and a good contribution to the literature. You do a good job of motivating your work and pointing out both where you expand on previous theory and where your approach has limitations. The modeling is a bit difficult to follow in places, with a lot of notation, but I think you have done as good a job of clarifying things as one could hope for – perhaps one other useful feature that you could add is a table summarizing the mathematical notation, including precise explanations of the use subscripts, superscripts and all the various accents.

As a whole I have little to add on top of what previous reviewers have already said. Your responses to their concerns were all thorough and satisfactory, and so my comments below are mainly just minor grammar and style fixes. In particular the additions of more background on previous work and more detail on the limitations of your model were big improvements to the manuscript. Despite its limitations this work provides a valuable step forward that should be of good general interest and I would be more than happy to see it published.

Specific comments:

Line 61: Is there an error in the wording “mortality risk at age associated with a life history”? Perhaps specify an age, and what does the “associated with a life history” refer to? Maybe change to just “mortality risk at a given age”?

Line 187: Change “another age class” -> “any age class”? You specify that individuals can be the same age, but “another” seems to imply that the same is excluded.

Eq. 6, 8, line 331: Does “ $f \cdot v$ ” refer to the dot product of f and v ? If so make sure to use centered dot instead of this period notation which confused me at first.

Line 298: “...given w_{yx} represents...” -> “...given w_{yx} , which represents...”

Line 307 & line 336: between age class -> between age classes?

Line 454: Perhaps change “accounts for transfers” -> “accounts for social transfers”, or even some more words reminding the readers how you used the word transfers. (Given the more summary-like tone at the beginning of the discussion and that it’s been a while since you introduced transfers.)

Line 500: “are results are” -> “our results are”

Line 504: Individuals reproducing and surviving exactly at population average rates is unlikely to hold for any N . The “especially at small N ” clause refers rather to when it matters most, so perhaps rewrite to “...which can have consequences especially at small N ” or something similar.

Line 554-555: I didn’t understand this argument. How would direct fitness benefits be incorporated by delaying the age at which returns from social interactions are realized?

Eq. A4: On the bottom line, shouldn’t the term $r_{\hat{}}(1)$ be $r_{\hat{}}(y-x)$, if the case is the opposite of on the top line (as you write on line 628).

Figure 2 legend, 2nd line from bottom: Problem with parentheses and punctuation, and your methods section is now called Methods and not Model. So, rewrite “...and $\omega = 50$. (see Model and SOM D) The relatedness...” -> “...and $\omega = 50$ (see Methods and SOM). D) The relatedness...”

Extended data figures 1-4 legends:

Specify what the other parameter values are (α , β , κ , ϕ), or at least where one can find them.

Line 182, 213, 246, 279: Remove extra “the” in front of “(i)”.

Line 182, 213: “see Fig. 1” should be “see Fig. 2”.

Line 246, 279: “see Fig. 2” should be “see Fig. 3”.

Line 183, 215, 248, 281: Make this last clause also numbered: “...(N), and (iii) the juvenile dispersal rate...”,

Reviewer #2 (Remarks to the Author):

Journal Questions

- Does the manuscript have technical or conceptual flaws that should prohibit its publication? If so, please provide details.

While the question the authors ask is important, they do not cite previous publications (e.g., van Tienderen 1995, Lande & Arnold 1983) that provide at least the working materials for to addressing the same problem of how to estimate selection on correlated vital rates reflecting indirect effects. They also do not present results in a manner that demonstrates the utility of their work. Their results graphs (1C, 2C) do not compare the force of selection with vs. without including indirect effects and they do not isolate selection on indirect effects. Also, they include relatedness, which is crucial for inclusive fitness, but they do not examine the effect of different patterns of relatedness beyond altering dispersal.

van Tienderen, Peter H. "Elasticities and the link between demographic and evolutionary dynamics." *Ecology* 81.3 (2000): 666-679.

Lande, Russell, and Stevan J. Arnold. "The measurement of selection on correlated characters." *Evolution* (1983): 1210-1226.

- Are the conclusions original? If not, please provide relevant references.

Although incorporating inclusive fitness into models of selection is a worthy goal and would improve our understanding of selection among social animals, the methods they present are complicated and unwieldy compared to existing methods for calculating "Integrated Sensitivities" (van Tienderen 1995), which also provide estimates of the force of selection on correlated vital rates. Although they do not cite van Tienderen (1995) or Lande & Arnold (1983), selection on correlated vital rates has been addressed by these authors and the "fractional copies" individuals contribute via fitness transfers are essentially looking at the same problem. If traits are correlated, then the methods presented by van Tienderen (1995) estimate the potential contributions of fitness transfers. Since the fitness transfers of Roper et al. are equivalent to the perturbations of other correlated vital rates when a focal vital rate changes, these Integrated Sensitivities could be used to estimate inclusive fitness under selection with much simpler notation.

- Do you feel that the results presented are of immediate relevance for people in your own discipline or for a broader audience? If you recommend publication, please outline briefly what you consider to be the outstanding features.

Addressing inclusive fitness in models of selection would be of interest to many in the field, but the unnecessarily complicated math and the large number of novel variables that would, in practice, be hard or impossible to estimate in most cases, make it difficult to see how these methods would be useful to the broad array of researchers that might apply them. At the very least, I would suggest reducing the number of variables and including those is variable definitions table, and providing a simpler application of the basic model to show how it can provide additional information beyond classic measures of the force of selection.

- If you feel that specific additional experiments would strengthen the case for publication in *Communications Biology*, please provide suggestions.

Notes on Integrated Sensitivities

Incorporating inclusive fitness into selection is an important problem and helps to explain life histories

that defy expectations based on Hamilton's selection. However, previously published and much simpler methods suffice to answer the same question. In particular, van Tienderen's (1995) Integrated Sensitivities incorporate indirect fitness effects of particular vital rates (e.g., juvenile survival or adult fertility) that are made through correlations with other vital rates.

Here, $IS_i = \sum_j r_{ij} s_j \sqrt{V_j/V_i}$ [eq 3]

where r_{ij} is the correlation ($\text{corr}(x_i, x_j)$), s_j is the sensitivity of vital rate x_j and V_j/V_i is the ratio of the variance of x_j and x_i . Note that lower-level vital rates (x_j, x_j) are a vectorized list of nonzero matrix elements ($A = \{a_{ij}\}$) or components of matrix elements, which in an age-structured model represent either survival ($p_i = a_{i,i+1}$) or fertility ($b_i = a_{1i}$) of age i individuals. IS_i , then, reflects the force of selection on vital rate (x_i) after incorporating indirect effects that drive vital rate correlations (e.g., negative correlations between juvenile survival and adult fertility reflecting survival costs of reproduction). Since Roper et al. use r_{ij} for the relatedness between individuals, the notation below uses p_{ij} for $\text{corr}(x_i, x_j)$ instead of r_{ij} .

In the absence of stochasticity, $V_i = V_j = 0$ so [eq 3] reduces to

$IS_i = \sum_j \rho_{ij} s_j$, and in the absence of correlations, reduces to classic fitness sensitivities $IS_i = s_i$.

IS_i is in the currency of fitness (λ). In the classic case, $\lambda = \sum_j s_i x_i$, where x_i are vital rates (e.g., age-specific survival or fertility).

In the case here,

$$\lambda = \sum_i IS_i x_i = \sum_j \sum_j \rho_{ij} s_j x_i$$

s_i is the fitness sensitivity ($s_i = d\lambda / d x_i$) and x_i is the focal vital rate.

To translate this into inclusive fitness effects, we can scale by mean relatedness (r_{ij}) between ages i and j .

$$\lambda = \sum_i IS_i x_i = \sum_j \sum_j r_{ij} \rho_{ij} s_{ij} a_{ij}$$

While it would be very useful to extend van Tienderen's (1995) approach to incorporate inclusive fitness through vital rate correlations (reflected in Roper et al.'s fractional individual transfers), Roper et al. attempt to reinvent the wheel here and introduce a large number of new variables that unnecessarily complicate the equations.

Responses to reviewers are in blue font, and are labelled numerically in the format **RX**.

Responses to R1

Reviewer #1 (Remarks to the Author):

General feedback:

Thank you for this nice manuscript, which I found to be very well written and a good contribution to the literature. You do a good job of motivating your work and pointing out both where you expand on previous theory and where your approach has limitations. The modeling is a bit difficult to follow in places, with a lot of notation, but I think you have done as good a job of clarifying things as one could hope for – perhaps one other useful feature that you could add is a table summarizing the mathematical notation, including precise explanations of the use subscripts, superscripts and all the various accents.

As a whole I have little to add on top of what previous reviewers have already said. Your responses to their concerns were all thorough and satisfactory, and so my comments below are mainly just minor grammar and style fixes. In particular the additions of more background on previous work and more detail on the limitations of your model were big improvements to the manuscript. Despite its limitations this work provides a valuable step forward that should be of good general interest and I would be more than happy to see it published.

R1: We thank the reviewer for taking the time to review our manuscript and for providing constructive feedback on our work. Addressing their key concern, we have added a new table (Table 1, pg 49) that summarises the mathematical notation, including precise explanations of subscripts, superscript, and various accents.

Specific comments:

Line 61: Is there an error in the wording “mortality risk at age associated with a life history”? Perhaps specify an age, and what does the “associated with a life history” refer to? Maybe change to just “mortality risk at a given age”?

R2: This was a mistake. We thank the reviewer for pointing it out. Changed to “mortality risk at age x ” (L61).

Line 187: Change “another age class” -> “any age class”? You specify that individuals can be the same age, but “another” seems to imply that the same is excluded.

R3: We have re-written this sentence to clarify that transfers are between any age classes, including individuals of the same age (L187-L188). We have also included a new paragraph that better details each inclusive fitness contribution in [1], including transfers between same aged individuals (L256-L275).

(L187-L188) Here, instead, we define transfers between age classes as the per capita net contribution of all helpful or harmful social behaviours.

Eq. 6, 8, line 331: Does “ $f \cdot v$ ” refer to the dot product of f and v ? If so make sure to use centered dot instead of this period notation which confused me at first.

R4: Yes, it does - we have now fixed this point. Eq. 6 & 8.

Line 298: "...given w_{yx} represents..." -> "...given w_{yx} , which represents..."

R5: Changed (L308-L311).

(L308-L311) Finally, given w_{yx} , which represents the class y offspring of a class x individual (genetic offspring equivalents), dw_{yx} then represents the difference in the contribution of a mutant individual aged x to individuals aged y relative to the resident population.

Line 307 & line 336: between age class -> between age classes?

R6: Fixed (L317 & L345).

Line 454: Perhaps change "accounts for transfers" -> "accounts for social transfers", or even some more words reminding the readers how you used the word transfers. (Given the more summary-like tone at the beginning of the discussion and that it's been a while since you introduced transfers.)

R7: Changed (L478-L481).

(L478-L481) First, the force of selection acting on the survival rate of that age class is the product of future inclusive reproductive value (IRV), rather than conventional reproduction value (RV^{38}), and a modified version of the asymptotic frequency (stationary age distribution) of that age class that accounts for social transfers.

Line 500: "are results are" -> "our results are"

R8: Fixed (L528).

(L528)....highlight the extent to which our results are...

Line 504: Individuals reproducing and surviving exactly at population average rates is unlikely to hold for any N . The "especially at small N " clause refers rather to when it matters most, so perhaps rewrite to "...which can have consequences especially at small N " or something similar.

R9: Re written this section to clarify (L532-L534).

(L532-L534) In reality, demographic stochasticity is likely to have important consequences on population structure and relatedness, impacting inclusive fitness forces of selection, especially at a small value of N

Line 554-555: I didn't understand this argument. How would direct fitness benefits be incorporated by delaying the age at which returns from social interactions are realized?

R10: We have clarified here that we were referring to group augmentation theory, which posits that future benefits to individual survival/reproduction can select for alloparental care, irrespective of kinship. To model such benefits, the social transfer can be modified so that it benefits the actor's direct fitness in the future, in addition to increasing the recipient's direct fitness (which will yield indirect fitness, when actor and recipient are related) (L580-L584).

(L580-L584) In many cooperative breeding systems, however, direct fitness returns from social interactions can also explain alloparental care³¹. In such systems, an individual's social transfers would ultimately benefit its own future survival or reproduction, (as well as its indirect fitness, where recipients of help are kin), as hypothesised by group augmentation theory⁷⁴.

Eq. A4: On the bottom line, shouldn't the term $\hat{r}(1)$ be $\hat{r}(y-x)$, if the case is the opposite of on the top line (as you write on line 628).

R11: This is not the case, but it is our fault for not describing correctly this term correctly. Because the individual is aged 1 at $y-x$, the relatedness is $\hat{r}(1)$, to an older individual aged y (that is not the individual's mother). We have re-written this line to clarify the correct definition for the term (L659-L666). Also, we noted from the previous manuscript that we had used f rather than s for asymptotic frequency (now L670-L671), which we have now changed in Appendix A. This does not change any of the calculations.

(L659-L666) The final scenario (bottom row of [A4]) considers the case when the individual aged y is older than the individual aged x . In this case the logic is similar to the case when the individual aged x is older than the individual aged y . With probability $h(y-x)$, the individual aged x is the offspring of an individual aged y , and so relatedness is one. With probability $k(y-x)$, the individual aged x is the offspring of another individual on the patch at time $y-x$, when the individual aged x was 1. Therefore, the relatedness to the individual aged y is equal to the average relatedness of a newborn that doesn't disperse upon establishing ($\hat{r}(1)$). With probability $(1-h(y-x)-k(y-x))$ the individual aged x dispersed from elsewhere in the population and so relatedness is equal to 0.

(L670-L671) we can do this by weighting each age class specific relatedness term by the asymptotic frequencies (s_x) of the relevant age classes:

Figure 2 legend, 2nd line from bottom: Problem with parentheses and punctuation, and your methods section is now called Methods and not Model. So, rewrite "...and $\omega = 50$. (see Model and SOM D) The relatedness..." -> "...and $\omega = 50$ (see Methods and SOM). D) The relatedness..."

R12: We have re-written the figure legend for Figure 2 given the new figure structure, and also fixed these errors (pg. 46).

Extended data figures 1-4 legends:

Specify what the other parameter values are (α , β , κ , ϕ), or at least where one can find them.

Line 182, 213, 246, 279: Remove extra "the" in front of "(i)".

Line 182, 213: “see Fig. 1” should be “see Fig. 2”.

Line 246, 279: “see Fig. 2” should be “see Fig. 3”.

Line 183, 215, 248, 281: Make this last clause also numbered: “...(N), and (iii) the juvenile dispersal rate...”,

R13: We have fixed these issues in the Extended Data Figure Legends (see SOM_Revised File).

Responses to R2

Reviewer #2 (Remarks to the Author):

Journal Questions

- Does the manuscript have technical or conceptual flaws that should prohibit its publication? If so, please provide details.

While the question the authors ask is important, they do not cite previous publications (e.g., van Tienderen 1995, Lande & Arnold 1983) that provide at least the working materials for to addressing the same problem of how to estimate selection on correlated vital rates reflecting indirect effects. They also do not present results in a manner that demonstrates the utility of their work. Their results graphs (1C, 2C) do not compare the force of selection with vs. without including indirect effects and they do not isolate selection on indirect effects. Also, they include relatedness, which is crucial for inclusive fitness, but they do not examine the effect of different patterns of relatedness beyond altering dispersal.

van Tienderen, Peter H. "Elasticities and the link between demographic and evolutionary dynamics." *Ecology* 81.3 (2000): 666-679.

Lande, Russell, and Stevan J. Arnold. "The measurement of selection on correlated characters." *Evolution* (1983): 1210-1226.

R14: We thank the reviewer for taking the time to review our manuscript and for providing constructive feedback on our work. We have split our response to this section of their comments into three parts, addressing each of the reviewer’s concerns in turn.

Concern 1: *‘they do not cite previous publications...’*

We thank the reviewer for raising our attention to these publications. However, it is assumed in our model that there are no genetic correlations between survival, reproduction, and social effects. Although this assumption is a simplification, it allows for the independent sensitivity analysis, following Hamilton (1966 *J Theor Biol*), Caswell (1978 *Theor. Popul. Biol*), and specifically for our model, Ronce & Promislow (2010 *Proc R Soc B*). To properly parameterize the model with the actual relationships/trade-offs between these vital rates would require an exhaustive experimental manipulation of a system. This action is well outside of the modelling nature of this work. However, we have addressed the lack of genetic correlations in the Discussion (L546-L549), and note the worthy future analysis of genetic correlations between traits. Second, from our understanding of the papers, and the reviewers comments, although ‘relatedness’ can be added to integrated sensitivities, alone the

methodology cannot be used to quantify inclusive fitness by stripping away social effects to the individual, and augmenting their social effects on others. This methodology is key for our analysis, and was the reason we chose to extend Ronce & Promislow’s methodology to do so. We have made this methodology clearer by editing paragraph L223-L240.

Concern 2: ‘They also do not present results in a manner that demonstrates the utility of their work’

We agree with the reviewer that the comparison of our inclusive fitness forces of selection with standard forces of selection would be useful in displaying the utility of our work. We have revised our figures and figure legends (Fig. 2 pg. 45-46, Fig. 3 pg. 47-48) to display the differences in quantifying the forces of selection in an age-structured population using different indicators of selection. We have added text (L411-L425) to introduce this comparison.

(L410-L425) To allow comparison our results with Hamilton’s² forces of selection, we define $m(x)$ as the age-specific effective fecundity in a population with complete dispersal ($d = 1$), no transfers, and the same age structure as our population¹⁴:

$$m(x) = b(x) \frac{N(1 - \bar{p})}{N\bar{b}}$$

[15]

Hamilton’s forces of selection can then be computed for age-specific survival ($p(x)$) and fecundity ($m(x)$) in a stationary population of the same age-structure as our population (described by A), were vital rates have not been translated into inclusive fitness contributions., The force of selection acting on survival is then $\frac{\sum_{y=x+1}^{\omega} l(y)m(y)}{T}$ and for reproduction is $\frac{l(x)}{T}$, with $T = \sum_{y=1}^{\omega} l(y)m(y)y$ giving the mean generation time^{2,14}. Fig. 2C and Fig. 3C then show the forces of selection acting on survival and reproduction at age x using Hamilton’s indicators, compared to our forces of selection which take juvenile dispersal and transfers into account ([8] and [10]).

Concern 3: ‘they do not examine the effect of different patterns of relatedness beyond altering dispersal’

We politely disagree with this comment, as we do in fact consider effects of dispersal, age-specific survival, age-specific reproduction, and patch population size on patterns of relatedness.

Our calculations for relatedness (r) follow Taylor & Irwin (2000 Evolution) to determine r in an age-structured infinite island population model. Dispersal is key to include in our model as it is the only way to create the variation in relatedness to simulate kin competition, under the assumption of asexuality. However, r is also driven by the patterns of age-specific survival and reproduction, and also the patch population size, N . These components can all contribute

importantly to r , shown in Appendix A. For example, a lower N , would, all else being equal increase average r to the patch. This is because the $\frac{1}{N}$ term, displaying the r of 1 of an individual to themselves, would carry more weight. Additionally, if age-classes aren't producing offspring (e.g. juveniles), all else being equal, their average relatedness to the patch will decrease. These terms all interact to determine age-specific r . We have added a sentence in 'Inclusive Fitness Model' to make this clearer (L165-L168).

(L165-L168) *A proportion of offspring of age class 1 disperse to other patches; when this proportion is between 0 and 1, patches comprise both kin and non-kin, leading to variation in pairwise relatedness. Variation in pairwise relatedness is also driven by the number of individuals on the patch, and age-specific rates of reproduction and survival (see Appendix A).*

References for the above comments

1. Hamilton, W. D. The moulding of senescence by natural selection. *J. Theor. Biol.* **12**, 12–45 (1966).
2. Caswell, H. A general formula for the sensitivity of population growth rate to changes in life history parameters. *Theor. Popul. Biol.* **14**, 215-230 (1978).
3. Ronce, O. & Promislow, D. Kin competition, natal dispersal and the moulding of senescence by natural selection. *Proc. R. Soc. B.* **277**, 3659-3667 (2010).
4. Taylor, P.D. & Irwin, A.J. Overlapping generations can promote altruistic behavior. *Evolution.* **54**, 1135–1141 (2000).

- Are the conclusions original? If not, please provide relevant references.

Although incorporating inclusive fitness into models of selection is a worthy goal and would improve our understanding of selection among social animals, the methods they present are complicated and unwieldy compared to existing methods for calculating “Integrated Sensitivities” (van Tienderen 1995), which also provide estimates of the force of selection on correlated vital rates. Although they do not cite van Tienderen (1995) or Lande & Arnold (1983), selection on correlated vital rates has been addressed by these authors and the “fractional copies” individuals contribute via fitness transfers are essentially looking at the same problem. If traits are correlated, then the methods presented by van Tienderen (1995) estimate the potential contributions of fitness transfers. Since the fitness transfers of Roper et al. are equivalent to the perturbations of other correlated vital rates when a focal vital rate changes, these Integrated Sensitivities could be used to estimate inclusive fitness under selection with much simpler notation.

R15: We have addressed our general motivations for our methods in R14, as well as the key point that there is no requirement that the traits under study be correlated. Social transfers represent an independent vital rate in the same currency as survival and reproduction. Additionally, we hope that the new Table 1 (pg. 49) makes clear the distinction between parameters required for the general analysis and numerical solutions. For example, the Siler model [12] and reproduction model [14]) aren't required for general solutions.

- Do you feel that the results presented are of immediate relevance for people in your own discipline or for a broader audience? If you recommend publication, please outline briefly what you consider to be the outstanding features.

Addressing inclusive fitness in models of selection would be of interest to many in the field, but the unnecessarily complicated math and the large number of novel variables that would, in practice, be hard or impossible to estimate in most cases, make it difficult to see how these methods would be useful to the broad array of researchers that might apply them. At the very least, I would suggest reducing the number of variables and including those is variable definitions table, and providing a simpler application of the basic model to show how it can provide additional information beyond classic measures of the force of selection.

R16: To address the concerns of both reviewers, we have added a table of notation (see Table 1; pg 49). We have addressed the concern of our methodology in previous comments (R14 & R15). As we have been clear, our accounting methodology is required for calculating inclusive fitness accurately, rather than just adding relatedness in.

We believe our framework can be built on for both future theoretical and practical work. Our discussion proposes extensions, addresses the limitations of the model, and also speculating on further possible theoretical treatments. In our previous comments and the new Table 1, we have also aimed to clarify the difference between general and numerical applications of our model, which limits the number of parameters required for general applications.

We are currently engaged in analyses of senescence rates and lifespan in cooperative breeding species, with which we aim to test predictions from our model. Literature searches to support these analyses confirms that information on the main variables from our model is known for a range of species. Demographic rates (age-specific survival and reproduction), average group size, dispersal, reproductive skew (generating differences in the proportion of offspring in a group born to certain individuals) are all well studied. These variables could be used to quantify relatedness; however, microsatellite estimates of relatedness are also widely available for many social species, which could be added directly. In fact, for some species, age-specific relatedness has been estimated (e.g. Ellis *et al.* 2022 Nat Eco Evo).

We agree with the reviewer that the final variables required for a full general exploration, age-specific social transfers, may be difficult to estimate, but general effects of helpers in cooperative breeding systems are well established (see e.g. Downing *et al.* 2021 Proc R Soc B). For example, when juvenile helpers have an average effect on breeder reproductive success, this could be used to parameterise the model (albeit not completely age-explicitly). We also used estimates of the helping effect of post-reproductive adults on the survival of juveniles in killer whales (Nattrass *et al.* 2019 PNAS), to help parameterise one of our examples (Figure 2). We hope our work stimulates researchers on longitudinal studies to consider measuring age-specific helping rates.

Finally, to provide the simpler application of our model, we refer the reviewer and editor back to our responses in R14, and the new Figures 2 and 3.

References for the above comments

1. Ellis, S. *et al.* Patterns and consequences of age-linked change in local relatedness in animal societies. *Nat Eco. Evol.* **6**, 1766-1776 (2022).
2. Downing, P.A. Griffin, A.S. & Cornwallis, C.K. Hard-working helpers contribute to long breeder lifespans in cooperative birds. *Proceedings of the Royal Society B: Biological Sciences*, 376 20190742 (2021).
3. Natrass, S. *et al.* Postreproductive killer whale grandmothers improve survival of their grandoffspring. *Proc. Natl Acad. Sci. USA.* **116**, 26669-26673 (2019).

- If you feel that specific additional experiments would strengthen the case for publication in Communications Biology, please provide suggestions.

Notes on Integrated Sensitivities

Incorporating inclusive fitness into selection is an important problem and helps to explain life histories that defy expectations based on Hamilton's selection. However, previously published and much simpler methods suffice to answer the same question. In particular, van Tienderen's (1995) Integrated Sensitivities incorporate indirect fitness effects of particular vital rates (e.g., juvenile survival or adult fertility) that are made through correlations with other vital rates.

Here, $IS_i = \sum_j r_{ij} s_j \sqrt{V_j/V_i}$ [eq 3]

where r_{ij} is the correlation ($\text{corr}(x_i, x_j)$), s_j is the sensitivity of vital rate x_j and V_j/V_i is the ratio of the variance of x_j and x_i . Note that lower-level vital rates (x_j, x_i) are a vectorized list of nonzero matrix elements ($A = \{a_{ij}\}$) or components of matrix elements, which in an age-structured model represent either survival ($p_i = a_{i,i+1}$) or fertility ($b_i = a_{1i}$) of age i individuals. IS_i , then, reflects the force of selection on vital rate (x_i) after incorporating indirect effects that drive vital rate correlations (e.g., negative correlations between juvenile survival and adult fertility reflecting survival costs of reproduction). Since Roper et al. use r_{ij} for the relatedness between individuals, the notation below uses ρ_{ij} for $\text{corr}(x_i, x_j)$ instead of r_{ij} .

In the absence of stochasticity, $V_i = V_j = 0$ so [eq 3] reduces to

$IS_i = \sum_j \rho_{ij} s_j$, and in the absence of correlations, reduces to classic fitness sensitivities $IS_i = s_i$.

IS_i is in the currency of fitness (λ). In the classic case, $\lambda = \sum_j s_i x_i$, where x_i are vital rates (e.g., age-specific survival or fertility).

In the case here,

$$\lambda = \sum_i IS_i x_i = \sum_j \sum_j \rho_{ij} s_{ij} x_i$$

s_i is the fitness sensitivity ($s_i = d\lambda / dx_i$) and x_i is the focal vital rate.

To translate this into inclusive fitness effects, we can scale by mean relatedness (r_{ij}) between ages i and j .

$$\lambda = \sum_i IS_i x_i = \sum_j \sum_j r_{ij} \rho_{ij} s_{ij} a_{ij}$$

While it would be very useful to extend van Tienderen's (1995) approach to incorporate inclusive fitness through vital rate correlations (reflected in Roper et al.'s fractional individual transfers), Roper et al. attempt to reinvent the wheel here and introduce a large number of new variables that unnecessarily complicate the equations.

R17: We thank the reviewer for supplying this information, which we have reviewed in detail. However, we refer the reviewer to our previous comments (R14, R15, and R16). We think that vital rate correlations, for example exploring trade-offs between survival and reproduction, are a worthy extension to our current work, but outside of the scope of this paper, which is the introduction of our initial framework (See R14 above). As the reviewer notes, the framework is already complex, and adding trade-offs at this stage would require further parameterisation. We also refer the editor and reviewer to our comments above on defining inclusive fitness accurately (R14).

REVIEWERS' COMMENTS:

Reviewer #1 (Remarks to the Author):

These edits have improved the paper significantly, and you have done a nice job of addressing both my and reviewer 2's comments from the previous round. Showing the comparison of forces of selection with and without age-specific social transfers is a nice addition. I now have no further hesitations in recommending this manuscript for publication, once the minor points are cleaned up.
Line 409: Syntax error - "To allow comparison our results..." -> either "To allow comparison of our results..." or "To allow comparing..."

Line 418: "were vital rates" -> "where vital rates"

Table 1: The explanation for $T^{x,yz}$ isn't super clear. Say something about that it represents the transfers? Maybe something like "The number of class y offspring contributed per capita through social transfers by age class x towards age class z, through the survival ($y = z + 1$) or reproduction ($y = 1$) of age class z individuals."

Responses to reviewers are in blue font, and are labelled numerically in the format **RX**.

Responses to R1

Reviewer #1 (Remarks to the Author):

These edits have improved the paper significantly, and you have done a nice job of addressing both my and reviewer 2's comments from the previous round. Showing the comparison of forces of selection with and without age-specific social transfers is a nice addition. I now have no further hesitations in recommending this manuscript for publication, once the minor points are cleaned up.

R1: We thank the reviewer for taking the time to provide feedback on our manuscript twice, and greatly improve it. We have revised the manuscript with respect to their comments, detailed below.

Line 409: Syntax error - "To allow comparison our results..." -> either "To allow comparison of our results..." or "To allow comparing..."

R2: Changed to "To allow comparison of our results" (L408-L409).

Line 418: "were vital rates" -> "where vital rates"

R3: Fixed (L417).

Table 1: The explanation for T^x_{yz} isn't super clear. Say something about that it represents the transfers? Maybe something like "The number of class y offspring contributed per capita through social transfers by age class x towards age class z, through the survival ($y = z + 1$) or reproduction ($y = 1$) of age class z individuals."

R4: We agree with the reviewer. The table 1 explanation has been changed to "*The number of class y offspring contributed per capita through social transfers by age class x to age class z, through the survival ($y = z + 1$) or reproduction ($y = 1$) of age class z individuals. The term is weighted by relatedness of age class x individuals to age class z individuals.*"